# Mendelian randomization prioritizes abdominal adiposity as an independent causal factor for liver fat accumulation and cardiometabolic diseases

Eloi Gagnon [1], William Pelletier[1], Émilie Gobeil [1], Jérôme Bourgault [1], Hasanga D. Manikpurage [1], Ina Maltais-Payette[1,2], Erik Abner [3], Nele Taba[3,4], Tõnu Esko[3], Patricia L. Mitchell[1], Nooshin Ghodsian[1], Jean-Pierre Després[5], Marie-Claude Vohl[2,6], André Tchernof[1,2], Sébastien Thériault [1,7] & Benoit J. Arsenault[1,8] ✉

## Abstract

**Background** Observational studies have linked adiposity and especially abdominal adiposity to liver fat accumulation and non-alcoholic fatty liver disease. These traits are also associated with type 2 diabetes and coronary artery disease but the causal factor(s) underlying these associations remain unexplored.

**Methods** We used a multivariable Mendelian randomization study design to determine whether body mass index and waist circumference were causally associated with non-alcoholic fatty liver disease using publicly available genome-wide association study summary statistics of the UK Biobank ($n = 461,460$) and of non-alcoholic fatty liver disease (8434 cases and 770,180 control). A multivariable Mendelian randomization study design was also used to determine the respective causal contributions of waist circumference and liver fat ($n = 32,858$) to type 2 diabetes and coronary artery disease.

**Results** Using multivariable Mendelian randomization we show that waist circumference increase non-alcoholic fatty liver disease risk even when accounting for body mass index (odd ratio per 1-standard deviation increase = 2.35 95% CI = 1.31–4.22, $p = 4.2e{-}03$), but body mass index does not increase non-alcoholic fatty liver disease risk when accounting for waist circumference (0.86 95% CI = 0.54–1.38, $p = 5.4e{-}01$). In multivariable Mendelian randomization analyses accounting for liver fat, waist circumference remains strongly associated with both type 2 diabetes (3.27 95% CI = 2.89–3.69, $p = 3.8e{-}80$) and coronary artery disease (1.66 95% CI = 1.54–1.8, $p = 3.4e{-}37$).

**Conclusions** These results identify waist circumference as a strong, independent, and causal contributor to non-alcoholic fatty liver disease, type 2 diabetes and coronary artery disease, thereby highlighting the importance of assessing body fat distribution for the prediction and prevention of cardiometabolic diseases.

**Plain language summary**

Non-alcoholic fatty liver diseases (NAFLD) affects around 25% of adults worldwide. NAFLD occurs when fat accumulates in the liver. Individuals with an elevated body weight are at higher risk of accumulating liver fat and developing cardiometabolic diseases such as NAFLD. Here, we show that people who inherit an increased likelihood to store fat inside the abdomen are at higher risk of liver fat accumulation and cardiometabolic diseases such as NAFLD, type 2 diabetes, and coronary artery disease. Conversely, individuals who store fat outside the abdomen may be at lower risk of liver fat accumulation and NAFLD. Based on these results, we conclude that the amount of fat in the abdomen rather than total body weight may be a key risk factor for cardiometabolic diseases.

[1] Centre de recherche de l'Institut universitaire de cardiologie et de pneumologie de Québec, Québec, QC, Canada. [2] School of Nutrition, Université Laval, Québec, QC, Canada. [3] Estonian Genome Center, Institute of Genomics, University of Tartu, Riia 23b, Tartu 51010, Estonia. [4] Institute of Molecular and Cell Biology, University of Tartu, Riia 23, Tartu 51,010, Estonia. [5] VITAM – Centre de recherche en santé durable, Université Laval, Québec, QC, Canada. [6] Centre NUTRISS, Institut sur la nutrition et les aliments fonctionnels, Université Laval, Québec, QC, Canada. [7] Department of Molecular Biology, Medical Biochemistry and Pathology, Faculty of Medicine, Université Laval, Québec, QC, Canada. [8] Department of Medicine, Faculty of Medicine, Université Laval, Québec, QC, Canada. ✉email: benoit.arsenault@criucpq.ulaval.ca

Non-alcoholic fatty liver disease (NAFLD) is characterized by hepatic lipid accumulation ranging from steatosis (>5% of liver weight is lipids) to non-alcoholic steatohepatitis (NASH, presence of inflammation)[1]. Although liver steatosis may be relatively benign in most cases, more severe forms of NAFLD such as NASH and hepatic fibrosis can lead to liver cirrhosis and hepatocellular carcinoma. Approximately 25% of the adult population globally is affected by NAFLD with the prevalence rapidly increasing and potentially becoming the leading cause of liver failure in the United States by 2025[2,3]. Adiposity and body fat distribution are closely linked with NAFLD[4]. In observational studies such as the INSPIRE ME study, a large international imaging study using computed tomography, waist circumference was closely associated with liver fat accumulation independently of body mass index (BMI)[5].

Studies have also shown that both liver fat accumulation/ NAFLD and waist circumference are associated with CAD and T2D[6–9]. However, whether liver fat accumulation is a causal factor of CAD and T2D remains to be elucidated and, more importantly, whether or not agents aimed at targeting NAFLD will ultimately decrease the risk of either T2D or CAD is unknown. In a previous investigation, we showed a strong genetic correlation between NAFLD, waist circumference, T2D, and CAD[10]. However, little is known about the directionality of these relations and whether NAFLD lies in the causal pathway linking abdominal adiposity and T2D/CAD.

In order to gain insight about the causality and directionality of these associations, causal inference methods such as Mendelian randomization (MR) have been developed[11]. MR uses genetic variants (which are randomly distributed at meiosis) such as single-nucleotide polymorphisms (SNPs), as instruments to infer causality. This method is in many ways comparable to a randomized control trial in which participants are naturally randomized based on the presence or absence of genetic variants that influence traits of interest[11]. In previous MR studies, a body fat distribution pattern consistent with low peripheral/subcutaneous fat accumulation and high intra-abdominal fat accumulation as estimated by the waist-to-hip ratio (WHR) adjusted for BMI was strongly associated with T2D and CAD[12,13]. However, we do not know if similar associations exist for NAFLD.

Extensions of the MR design, such as bidirectional MR and multivariable MR (MVMR), help in clarifying causal relations. Bidirectional MR refers to an analysis where both traits are alternately evaluated as exposure and outcome. This method has the potential to remove reverse causation bias by asserting the directionality of the relationship[14]. Multivariable MR can be used when multiple genetic variants are associated with two or more exposures. It conditions the effects of the SNPs of each exposure together to assess the effect of each exposure independently on the outcome. This method allows to test for mediation when two exposures share genetic variants as if they had been adjusted for one another[15].

Here, we used a MVMR study design to investigate the respective causal contributions of adiposity (defined using BMI) and abdominal adiposity (defined using waist circumference and the waist-to-hip ratio adjusted for BMI [WHRadjBMI]) to liver fat accumulation and NAFLD. Second, using a similar strategy, we aimed to determine if abdominal adiposity and liver fat accumulation are independent causal risk factors for T2D and CAD. Taken together, our triangulation of MR methods identify waist circumference as a strong, independent, and causal contributor to NAFLD, type 2 diabetes, and coronary artery disease.

## Methods

**Study populations.** Information on the cohorts used in this MR framework is presented in Supplementary Data 1. Briefly, we combined data from publicly accessible GWAS summary statistics of European ancestry in a two-sample MR setting. BMI and waist circumference: The summary statistics of BMI and waist circumference were obtained from the UK Biobank from 461,460 and 462,166 individuals respectively. The GWAS was performed by the MRC IEU open GWAS project[16]. GWAS summary statistics from the GIANT consortium were also included to replicate the estimates obtained with the UK Biobank. These summary statistics for BMI were obtained from a meta-analysis of up to 125 GWAS for 339,224 European individuals[17]. Summary statistics for waist circumference were obtained from a meta-analysis of 232,101 individuals[18]. Measures of BMI and waist circumference were self-reported or measured in a laboratory or in a healthcare setting. Measures were corrected for age, age squared, sex, ancestry-based principal components, and study sites. The resulting residuals were inverse ranked normal transformed with standard deviation (SD) of 1. WHR adjusted for BMI: WHR adjusted for BMI was calculated as the ratio of waist and hip circumferences adjusted for BMI in 485,486 Europeans in the UK Biobank[19]. Measures of WHR and BMI were self-reported, measured in a laboratory or measured in a healthcare setting. Measures of WHRadjBMI were corrected for age, age squared, sex, principal components, and study site. The resulting residuals were transformed to approximate normality with SD of 1 using inverse normal scores. We also included GWAS summary statistics for WHRadjBMI from 210,088 Europeans from the GIANT consortium[18]. In that study, WHRadjBMI was adjusted for age, age-squared, study-specific covariates and then inverse ranked normal transformed prior to genome-wide analysis. NAFLD: We performed a GWAS meta-analysis for clinical diagnosis of NAFLD (8434 cases and 770,180 controls) of European ancestry from four cohorts, as previously described[10]. Briefly, we performed a fixed effect GWAS meta-analysis of The Electronic Medical Records and Genomics (eMERGE) network[20], the UK Biobank, the Estonian Biobank and FinnGen using the METAL package[21]. NAFLD was defined using electronic health record codes or hospital records. Logistic regression analysis was performed with adjustment for age, sex, genotyping site and the first three ancestries-based principal components. Liver Fat: GWAS summary statistics for liver fat were obtained from a GWAS of 32,858 white British participants from the UK Biobank[22]. Magnetic resonance scans were annotated by trained radiologists following a standard procedure. Using this training dataset, deep learning algorithms were then applied to estimate liver fat. The resulting dataset comprises 32,860 liver fat quantification. Liver fat was regressed using BOLT-LMM on gene carrier status, adjusted for genetic sex, age, age$^2$, the first 10 principal components of genetic ancestry, scaled scan date, scaled scan time, and study center as fixed effects and genetic relatedness as a random effects term. The resulting residuals were inverse normal transformed prior to GWAS. Coronary artery disease: GWAS summary statistics for CAD were obtained from a GWAS on 122,733 cases and 424,528 controls from CARDIo-GRAMplusC4D and UK Biobank[23]. Samples from CARDIo-GRAMplusC4D were drawn from a mixed population (Europeans, East Asian, South Asian, Hispanic and African American), with the majority (77%) of the participants from European ancestry. Case status was defined by CAD diagnosis, including myocardial infarction, acute coronary syndrome, chronic stable angina, or coronary stenosis. We also used a different dataset GWAS summary statistics from the CARDIo-GRAMplusC4D excluding UK Biobank (60,801 CAD cases and 123,504 controls)[24]. Type 2 diabetes: GWAS summary statistics for type 2 diabetes were obtained from the DIAbetes Genetics Replication and Meta-analysis (DIAGRAM) consortium and UK Biobank (74,124 cases/824,006 controls)[25]. Case status was

defined by electronic health records, self-reports, or laboratory derived clinical diagnostics of T2D. We also used a different dataset from the DIAGRAM consortium excluding UK Biobank (26,676 T2D case and 132,532 controls)[26].

Some of the study samples used to derive our study exposures and outcomes included summary statistics from the UK Biobank, which lead to sample overlap. In univariable MR, sample overlap will bias the estimated results towards the null only when weak instrument is present. In MVMR, the direction of the bias is unclear but will occur only in the presence of weak instrument bias[27]. We included in our primary MR analysis the UK Biobank to increase power and included sensitivity analysis excluding the UK Biobank to remove sample overlap. All GWAS summary statistics were publicly available and accessible through URL. For all included genetic association studies, all participants provided informed consent, and study protocols were approved by their respective local ethical committee. Ethical approval was not required to conduct this study as it only used anonymized GWAS summary statistics.

**Selection of genetic variants and variants harmonization**. For univariable MR analysis, we selected all genome-wide significant SNPs (p-value < 5e−8). We then ensured the independence of genetic instruments by clumping all neighboring SNPs in a 10 Mb window with a linkage disequilibrium $r2 < 0.001$ using the European 1000-genome LD reference panel. SNPs and relevant association statistics can be found for each exposure in Supplementary Data 2. For multivariable MR analyses, we first extracted all genetic instruments that were previously selected for univariable MR analysis. We then pooled these SNPs to the lowest p-value corresponding to any of the exposures, using the same parameter setting as the univariable MR ($r2 = 0.001$ window = 10 Mb). We also included results of two other sensitivity analysis approaches: (1) prioritizing variants with lowest p value for BMI; (2) prioritizing SNPs with lowest p value for waist circumference. When NAFLD was used as an exposure in MVMR, we pooled the combined list of SNPs by selecting the SNP with the lowest p-value for NALFD. This procedure was implemented to select a maximum number of strong genetic instruments, as fewer genetic instruments are available for NAFLD exposure. SNPs in a 2 Mb window of the *HLA*, *ABO*, and *APOE* genetic regions were excluded due to their complex genetic architecture and their widespread pleiotropy (in GRCh37 6:28909037-30913661, 9:135130951-137150617, and 19:44409011-46412650, respectively). Exclusion of pleiotropic genetic regions satisfies the exclusion restriction and the exchangeability assumptions of instrumental variable analyses and strengthen inference of MR analyses. Harmonization was performed by aligning the effect sizes of different studies on the same effect allele. All GWAS summary statistics were reported on the forward strand. When a particular SNP was not present in the outcome datasets, we used a proxy SNPs ($r2 > 0.6$) obtained using linkage disequilibrium matrix of European samples from the 1000 Genomes Project. Instrument strength was quantified using the F-statistic[28], and the variance explained was quantified using the r2[29]. We calculated r2 for each individual SNP. For binary exposures, we calculated r2 using equation 10 in Lee et al., 2012[30] used in the get_r_from_lor function in the *TwoSampleMR* package. We calculated the F statistics following the formula $F = (\frac{n-k-1}{k})(\frac{R2}{1-R2})$. Where $n$ is the sample size, $k$ is the number of instruments used and $R2$ is the sum of the individual $r2$ of each SNP. These statistics can be found in Supplementary Data 3.

**Statistical analyses**. For univariable primary MR analysis, we performed the inverse variance weighted (IVW) method with multiplicative random effects with a standard error correction for under dispersion[31]. MR must respect three core assumptions (relevance, independence, and exclusion restriction) for correct causal inference. MR estimates bias occurs if the genetic instruments influence several traits on different causal pathways. This phenomenon, referred to as horizontal pleiotropy, can be balanced by using multiple genetic variants combined with robust univariable MR methods[32]. To verify if pleiotropy likely influenced the primary univariable MR results, we performed 6 different robust MR analyses: MR Egger[33], the MR-Robust Adjusted Profile Score (MR-RAPS)[34], the contamination mixture[35], the weighted median, the weighted mode and the MR-PRESSO[36], each making a different assumption about the underlying nature of the pleiotropy. Consistent estimates across methods provide further confirmation about the nature of the causal links. All continuous exposure estimates were normalized and reported on a SD scale. For dichotomous traits (i.e., diseased status on NAFLD, T2D and CAD), odds ratios were reported. Univariable MR analyses were performed using the *TwoSampleMR* V.0.5.6 package[37].

For multivariable primary MR analysis, we conducted the IVW method[38]. The use of MVMR is analogous to the inclusion of measured covariates in a multivariate linear regression. MVMR uses a set of overlapping genetic instrument to estimate the direct effect of an exposure on an outcome. As robust MVMR analyses, we used the multivariable MR-Egger[39], the multivariable median method, and the multivariable MR-Lasso method[40]. Similar to robust univariable MR analyses, each method makes different assumptions about the underlying nature of the pleiotropy and consistent estimates give confidence in the robustness of the causal findings. Multivariable MR analyses were performed using the *MendelianRandomization* V.0.5.1 package[41]. Conditionnal F-statistics were calculated with formula developed by Sanderson et al., in the *MVMR* V.0.2.0 package[42]. Percentage of mediation was quantified using the formula $(1 - \frac{\theta_2}{\theta_t})$ Where $\theta_2$ is the direct effect estimated with IVW-MVMR and $\theta_t$ is the total effect estimated with univariable IVW-MR[43].

**Reporting summary**. Further information on research design is available in the Nature Research Reporting Summary linked to this article.

## Results

**Causal effect of general and abdominal adiposity to NAFLD and liver fat accumulation**. We first investigated the causal effect of adiposity (defined by BMI or waist circumference) on NAFLD using Inverse Variance Weighted (IVW)-MR and other robust analyses described in the "Methods" section. Results from all univariable MR methods (Fig. 1 and Supplementary Data 4) including Egger's intercept (Supplementary Data 5) suggest that BMI and waist circumference are both causally associated with NAFLD. Using 370 SNPs ($r^2 = 0.05$; F-statistic = 60), a one SD-higher waist circumference had an odds ratio (OR) of 1.98 (95% confidence interval [CI]: 1.73–2.27, $p = 6.6e−23$) for NAFLD. Using 449 SNPs ($r^2 = 0.06$; F-statistic = 68), a one SD higher BMI had an OR of 1.66 95% CI = 1.49–1.85, $p = 2.3e−20$. Similar associations were found when exposures were derived from the Genetic Investigation of Anthropometric Traits (GIANT) consortium (Supplementary Fig. 1) and when liver fat accumulation was used as the outcome (Fig. 1).

To evaluate the effect of body fat distribution, we investigated the association of WHRadjBMI with NAFLD and liver fat accumulation using multiple univariable MR methods. WHRadjBMI is associated with an elevated waistline and lower BMI. A high WHRadjBMI is a marker of preferential intra-abdominal/visceral adipose tissue accumulation[13]. Using an

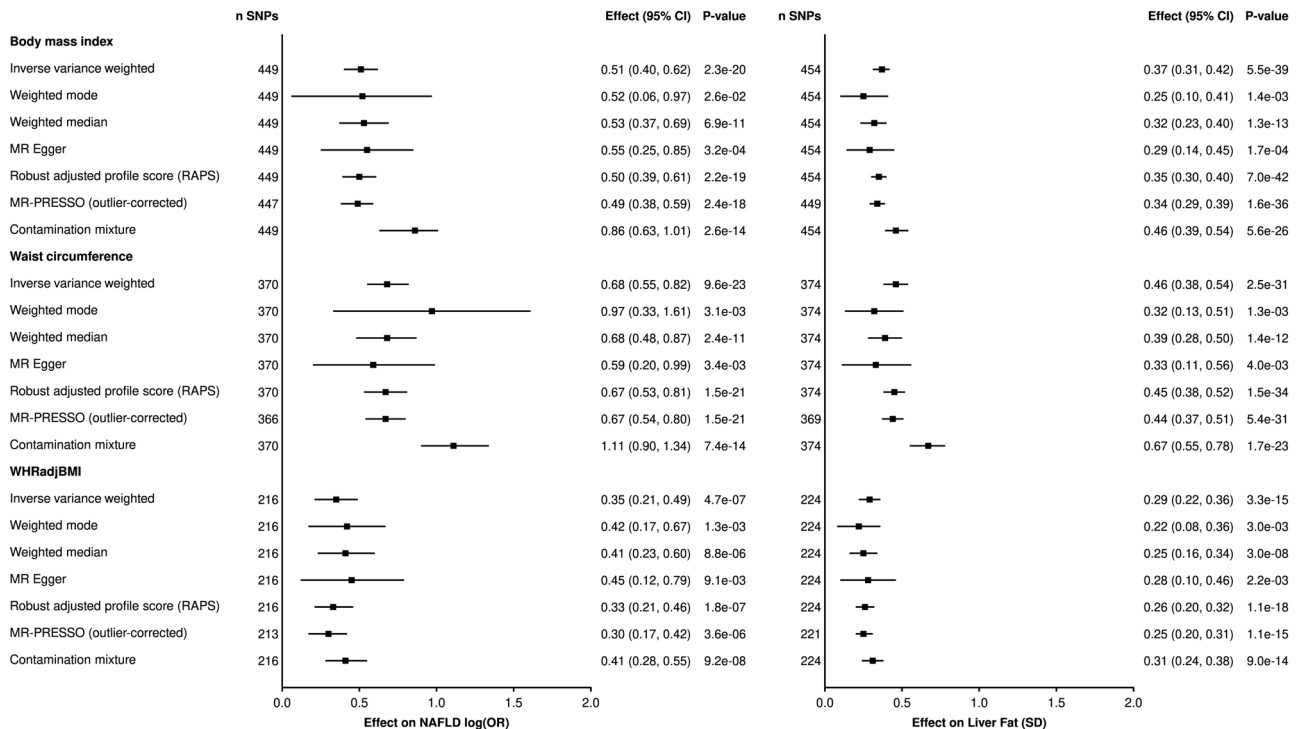

**Fig. 1 Causal effect of genetically-predicted anthropometric traits on non-alcoholic fatty liver disease (NAFLD) and liver fat accumulation.** Inverse-variance weighted Mendelian randomization (IVW-MR) and robust MR analyses were performed to assess the impact of one SD increase body mass index (BMI), waist circumference, and the waist-to-hip ratio adjusted for BMI on NAFLD and liver fat accumulation. Error bars are 95% confidence interval.

instrument of 216 SNPs ($r^2 = 0.03$, F-statistic = 74), a higher genetically predicted WHRadjBMI was associated with NAFLD across all univariable MR methods (OR = 1.42 95% CI = 1.24–1.62, $p = 4.7e{-}07$) (Fig. 1). Higher genetically predicted WHRadjBMI was also associated with liver fat accumulation across all univariable MR methods. Results were similar when deriving WHRadjBMI from GWAS summary statistics from GIANT. Altogether, these analyses provide evidence that body fat distribution patterns consistent with higher visceral fat accumulation is an important determinant of NAFLD.

To further investigate the impact of body fat distribution patterns on liver fat accumulation and NAFLD, we evaluated the direct causal effect of abdominal obesity irrespective of general adiposity in a MVMR analysis. BMI and waist circumference shared 114 instruments. When BMI and waist circumference were assessed together in MVMR, only waist circumference (2.35 95% CI = 1.31–4.23, $p = 4.2e{-}03$) retained a robust association with NAFLD. The effect of BMI on NAFLD upon adjustment for waist circumference was inconclusive (0.86 95% CI = 0.54–1.38, $p = 5.4e{-}01$) (Fig. 2). Conditional F-statistics for this MVMR analysis were low (1.54 and 1.55 for WC and BMI respectively). However, results were significant and consistent across all robust MVMR analyses and multivariable Egger intercept did not differ from zero indicating that pleiotropy is unlikely to have affected the results (Supplementary Data 6). Similar associations were found when liver fat accumulation was used as the outcome and when using GWAS summary statistics from the GIANT consortium as study exposures for waist circumference and BMI (Supplementary Fig. 2).

To confirm the impact of body fat distribution indices on liver fat accumulation and NAFLD, we investigated 159 adiposity-related genetic variants derived from ~322,154 subjects from the GIANT consortium[12]. These variants were previously classified into three groups based on the direction of their effects on BMI and WHR: those with positive ($p < 0.05$) association with BMI

and positive ($p < 0.05$) association with WHR (BMI+WHR+), negative ($p < 0.05$) association with WHR (BMI+WHR−) or null ($p > 0.05$) association with WHR (BMIonly+)[12]. Group-specific univariable MR using the 80 BMI+WHR+ instruments in the UK Biobank BMI revealed that BMI was positively associated with NALFD risk (OR = 1.61, 95% CI = 1.32–1.98, $p = 4.3e{-}06$). However, using the 24 BMI+WHR- SNPs, BMI was negatively associated with NAFLD (OR = 0.23, 95% CI = 0.09–0.56, $p = 1.4e{-}03$). Using the 25 BMIonly+ SNPs the effect of BMI on NAFLD was null (OR = 1.10, 95% CI = 0.66–1.86, $p = 7.1e{-}01$) (Fig. 3). These results were consistent in robust univariable MR analyses and when evaluating liver fat as the outcome and when using the GIANT consortium as exposure (Supplementary Data 7). Altogether, these results corroborate the univariable and MVMR analyses and provide additional support that intra-abdominal adiposity is a key driver of liver fat accumulation and NAFLD.

**Contributions of abdominal adiposity and liver fat to type 2 diabetes and coronary artery disease.** Since abdominal adiposity, NAFLD, T2D, and CAD are highly phenotypically correlated, we next explored the causal effect of abdominal adiposity and NAFLD/liver fat on cardiometabolic diseases. In univariable IVW-MR, using 374 SNPs ($r^2 = 0.05$; F-statistic = 60), a 1-SD increment in waist circumference increased T2D risk (OR = 3.65 95% CI = 3.25–4.1, $p = 1.8e{-}106$) (Fig. 4) and CAD risk (OR = 1.61 95% CI = 1.5–1.73, $p = 4.3e{-}40$) (Supplementary Data 4). Using 4 SNPs ($r^2 = 0.0005$; F-statistic = 2), there was evidence for causal effect of NAFLD on T2D, but not CAD (Supplementary Data 4). Since only four SNPs were associated with NAFLD at the genome-wide significance level ($p < 5e{-}8$), we investigated the relationship between NAFLD and T2D and CAD using a more lenient threshold ($p < 5e{-}6$). This analysis confirmed that genetically predicted NAFLD was associated with T2D but not

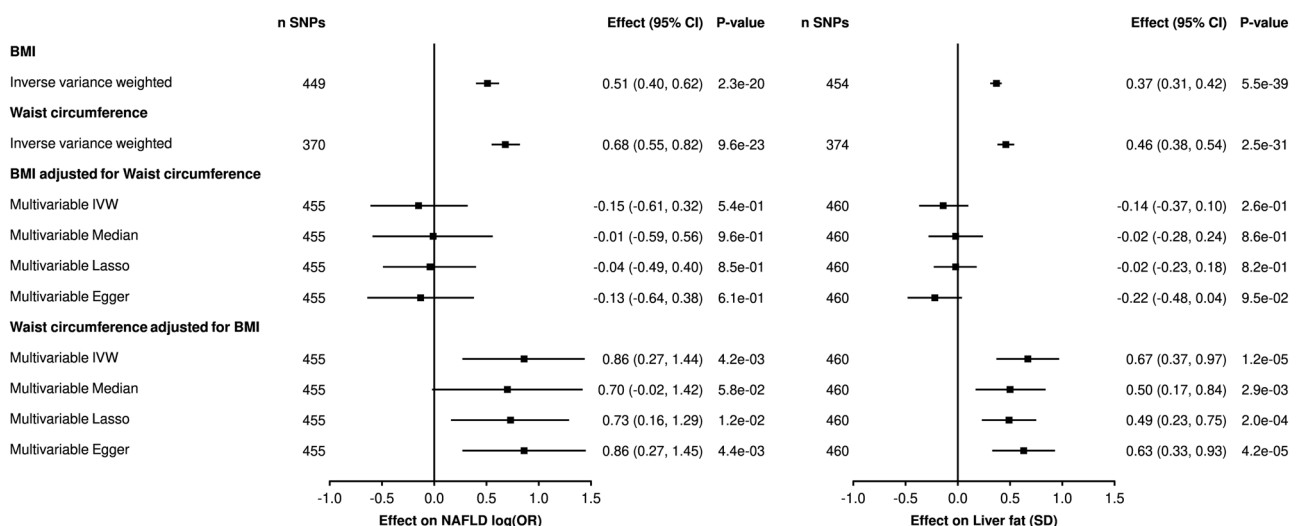

**Fig. 2 Causal effect genetically predicted waist circumference and body mass index (BMI) on non-alcoholic fatty liver disease (NAFLD) and liver fat accumulation using univariable and multivariable Mendelian randomization.** The association between waist circumference and body mass index (per 1-SD increase) and NAFLD and liver fat accumulation is presented using univariable inverse-variance weighted MR and multivariable MR using multiple robust methods. Error bars are 95% confidence interval.

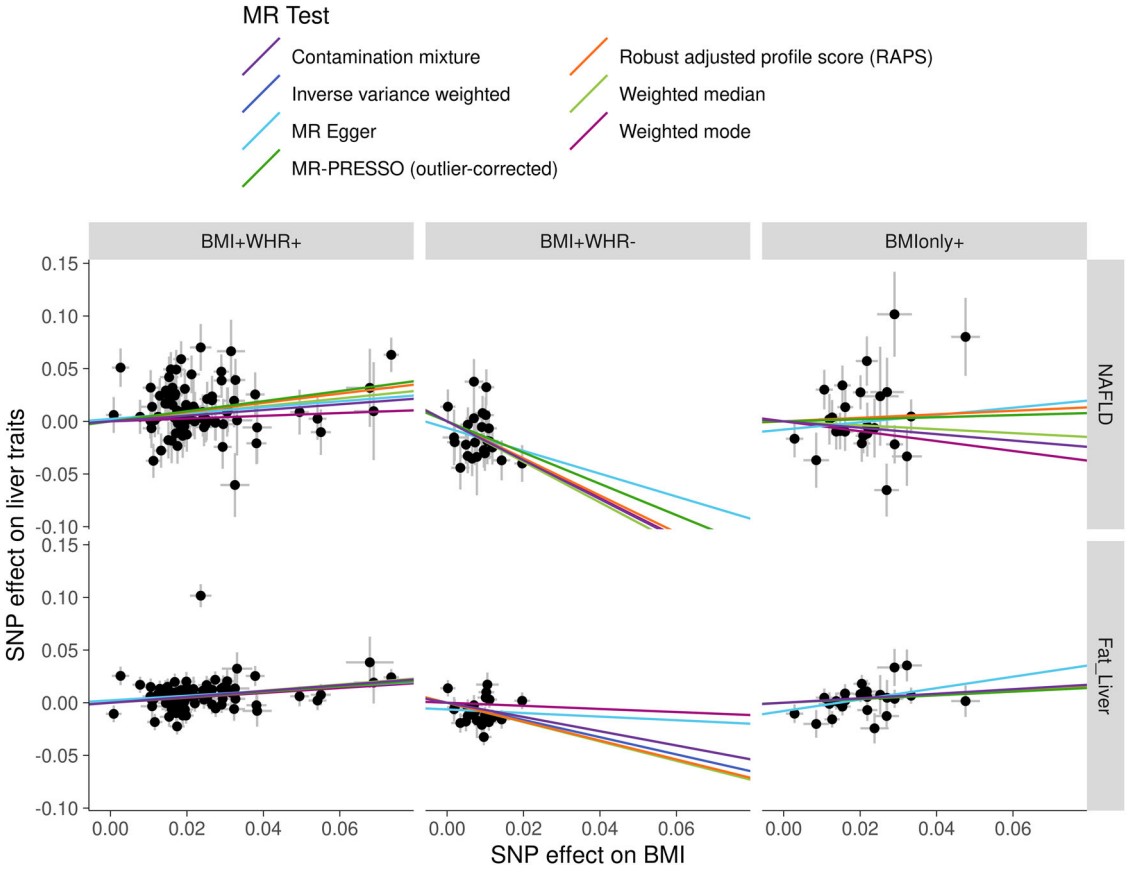

**Fig. 3 Effect of body mass index (BMI) variants on non-alcoholic fatty liver disease (NAFLD) and liver fat accumulation using group-specific Mendelian randomization.** A total of 159 genetic instruments categorized based on their association with BMI and waist-to-hip ratio (WHR): Left panel BMI+WHR+ (nominal significant effects on BMI and WHR with consistent directions), center panel BMI+WHR− (nominal significant effects on BMI and WHR with opposite directions) and right panel BMIonly+ (nominal significant effects on BMI only).

CAD (Supplementary Data 8–9). The binary factor NAFLD, which is diagnosed when liver fat percentage is above 5%, is akin to a dichotomization of the underlying continuous factor "liver fat". We therefore estimated the causal effect of the continuous variable "liver fat" on cardiometabolic diseases, as recommended[44]. Using 10 SNPs ($r^2 = 0.04$; F-statistic = 165), a 1-SD increase in liver fat increased the risk of T2D (OR = 1.26 95% CI = 1.08–1.47, $p = 3.8e{-}03$), but the effect on CAD was

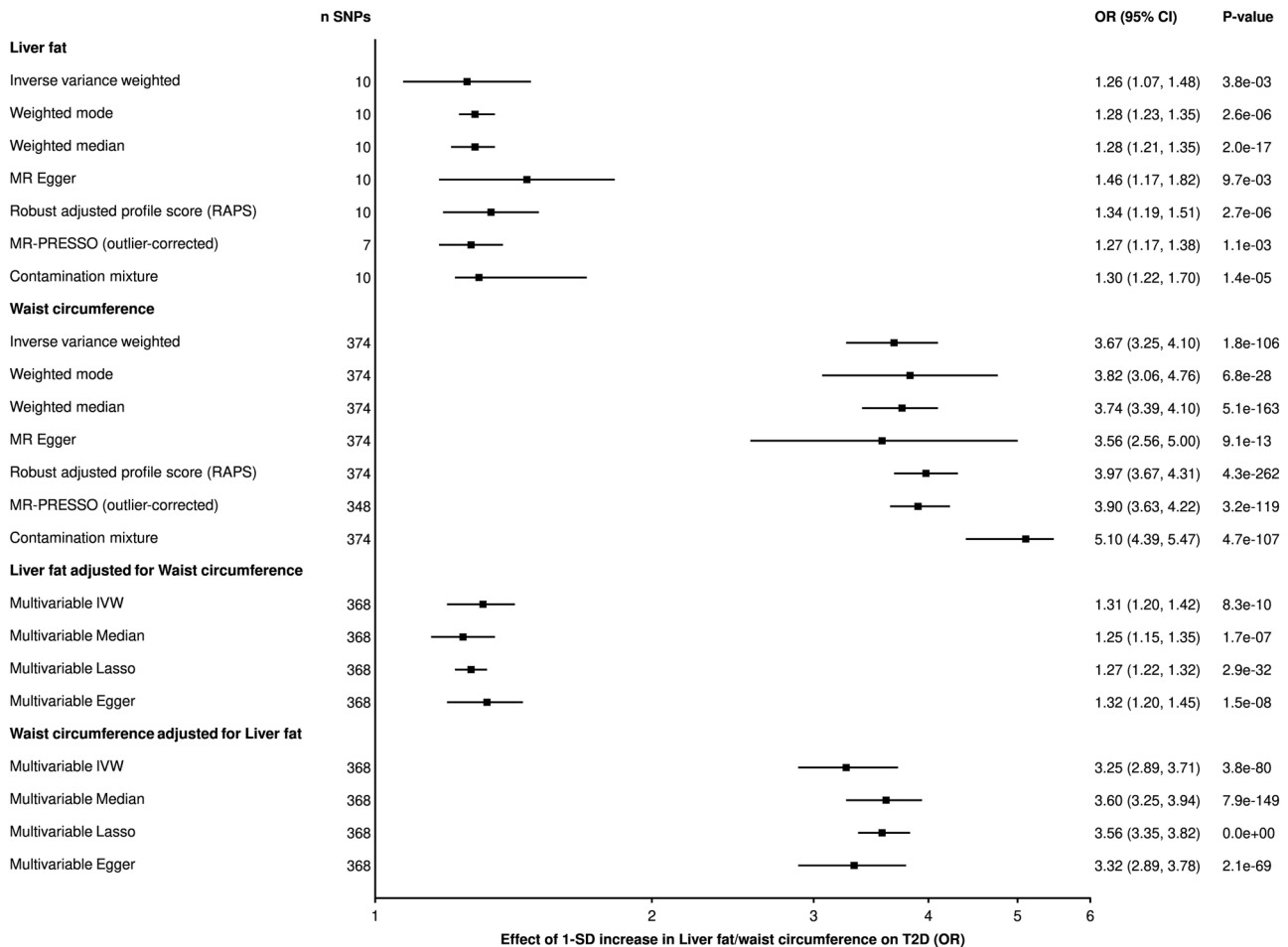

**Fig. 4 Causal effect of 1-SD increase of waist circumference and liver fat accumulation on type 2 diabetes (T2D) using univariable and multivariable Mendelian randomization (MR).** The effect of genetically-predicted waist circumference and liver fat accumulation on T2D using multiple robust MR methods is presented in the top panels and the effect of genetically-predicted waist circumference and liver fat accumulation on T2D using multiple robust multivariable MR methods is presented in the bottom panels. Error bars are 95% confidence interval.

inconclusive (OR = 0.90 95% CI = 0.75–1.10, $p = 3.0e{-}01$) (Fig. 4 and Supplementary Data 3). These causal effects were consistent for all robust univariable MR methods (Fig. 4). Of note, T2D increased liver fat accumulation and NAFLD while there was no evidence for an effect of CAD on liver fat accumulation and NAFLD (Supplementary Data 3).

Since our results have shown that waist circumference increased liver fat and that both traits increased the risk of T2D, we evaluated their respective causal contributions to T2D using MVMR. When waist circumference and liver fat were assessed together in MVMR, waist circumference (OR = 3.27 95% CI = 2.89–3.69, $p = 3.8e{-}80$) and liver fat (1.31 95% CI = 1.2–1.42, $p = 8.3e{-}10$) increased the risk of T2D. Results from robust MVMR methods were consistent with a causal effect of both waist circumference and liver fat on T2D (Fig. 4). Mediation analysis suggests that the impact of abdominal adiposity on T2D is 9% mediated by liver fat. In MVMR, the effect of waist circumference on T2D is 4.44 times larger than the effect of liver fat on T2D. The results were similar when deriving waist circumference instruments from GIANT (Supplementary Fig. 3) and when excluding the UK Biobank dataset from the outcome (Supplementary Data 4–5). Results of this analysis revealed that abdominal adiposity is a causal risk factor for CAD and T2D and that the effect of abdominal adiposity on T2D is only modestly (9%) mediated by liver fat.

## Discussion

In this study, we explored the relationships between general and abdominal adiposity and NAFLD using univariable and multivariable MR. We found that general and abdominal adiposity were causally linked to liver fat accumulation and NAFLD. Results of our multivariable MR analysis suggest that waist circumference is causally linked to liver fat accumulation and NAFLD regardless of BMI, while BMI is not causally linked with NAFLD once waist circumference is taken into account. Having established a causal role of abdominal adiposity on NAFLD and given the results of previous studies linking NAFLD to cardiometabolic diseases such as T2D[45,46] and CAD[47,48], we explored whether liver fat accumulation lies in the causal pathway linking abdominal adiposity to T2D and CAD. Using MVMR, our results support that the effect of abdominal adiposity on T2D is substantially larger than the effect of liver fat on T2D. We also showed that the association between abdominal adiposity and CAD is independent of liver fat, thereby highlighting the causal role of abdominal adiposity in the etiology of NAFLD, T2D, and CAD (Fig. 5).

Observational studies have documented similar associations[49]. Liu et al. used bidirectional MR to explore the relationship between NAFLD, adiposity, T2D, and lipid traits[46]. They found that both adiposity and abdominal adiposity had a causal effect on NAFLD. Our study provides additional support for a causal

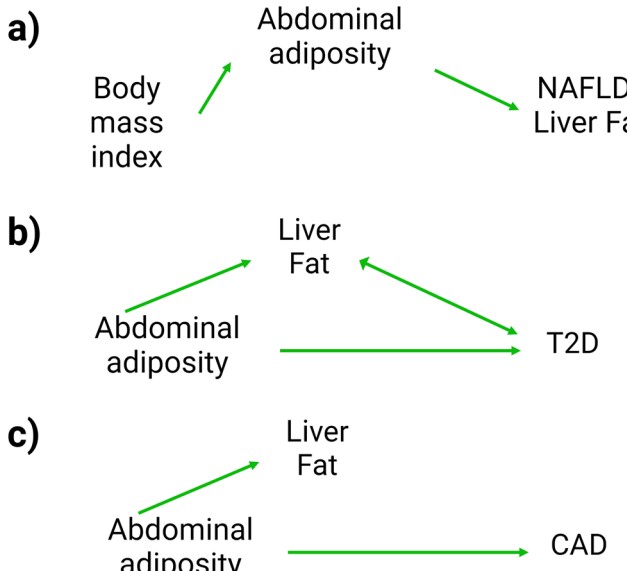

**Fig. 5 Schematic illustration of the main findings of the study. a** Both genetically-predicted body mass index and abdominal adiposity are associated with non-alcoholic fatty liver disease (NAFLD) and liver fat accumulation. However, the relationship between genetically-predicted BMI and NAFLD/liver fat accumulation is entirely mediated by genetically-predicted abdominal adiposity. **b** Genetically-predicted abdominal adiposity and NAFLD/liver fat accumulation are both associated with type 2 diabetes (T2D) and their associations with T2D are largely independent from one another. Genetically-predicted T2D is also associated with NAFLD/liver fat accumulation. **c** Genetically-predicted abdominal adiposity, but not genetically-predicted NAFLD/liver fat accumulation is associated with coronary artery disease (CAD). The association between genetically-predicted abdominal adiposity and CAD is not mediated by NAFLD/liver fat accumulation.

effect of abdominal adiposity on NAFLD using a larger study sample size for the study outcome (NAFLD) and multiple MR methods. Our study is, to our knowledge, the first to report using MVMR a causal association between abdominal adiposity and NAFLD that is independent of the BMI. In their MR investigation, Liu et al. also showed that NAFLD had a causal effect on T2D (OR: 1.3, 95% CI: 1.2, 1.4, $p = 8.3e{-}14$)[46]. They used 2 genetic instruments for NAFLD, making it impossible to perform pleiotropy robust MR analyses. In our analysis, NAFLD and liver fat was similarly associated with T2D and robust univariable MR analysis were consistent with a causal association. The association was slightly decreased when we accounted for abdominal adiposity using MVMR, suggesting that part of the effect of liver fat on T2D could be attributable to variants influencing primarily abdominal fat accumulation. On the other hand, the point estimate of waist circumference on T2D also only slightly decreased when we accounted for liver fat accumulation, suggesting that the effect of waist circumference on T2D is modestly mediated by liver fat accumulation.

The inability of subcutaneous fat to expand by hyperplasia may partly explain why visceral fat accumulation occurs in genetically predisposed individuals[50]. These excess lipids are then stored in lean tissues such as the liver, heart, and skeletal muscle promoting insulin resistance[4,51]. The mechanisms by which visceral fat contributes to NAFLD may also possibly be explained by the "portal vein theory"[52]. Visceral fat is mostly drained by the portal vein, which delivers its content to the liver and exposes it to high concentrations of free fatty acids and adipokines[53]. These have been hypothesized to lead to metabolic changes in the liver which

would ultimately lead to an increased production of VLDL particles, glucose, and inflammatory mediators as well as decreased insulin extraction, potentially leading to T2D and atherosclerosis[50,52].

From a clinical perspective, results of this study support the idea that previously reported associations between an elevated BMI and NAFLD may be explained by preferential abdominal fat accumulation reflected by higher waist circumference. Indeed, a significant number of individuals with elevated BMI have excess visceral fat increasing their risk of NAFLD[46,54,55]. Our results also underline the limitations of the sole use of BMI in clinical practice to assess the risk associated with obesity/ectopic fat distribution. The failure of BMI to capture cardiometabolic risk had already been suggested by observational and MR studies[4,5,56]. Our study adds evidence supporting waist circumference as a simple tool to assess obesity-related health hazards.

These results should encourage clinical interventions focused on visceral fat reduction, not only overall body weight reduction, to prevent cardiometabolic diseases such as NAFLD, T2D and CAD. Visceral fat can be targeted with physical activity and dietary interventions even in the absence of weight loss. A weight loss of about 5% can result in a 15–25% visceral fat reduction[57]. The Mediterranean diet as well as diets lower in fat and/or carbohydrate may be effective ways of reducing visceral fat, especially in physically active individuals[4,58,59]. There is also evidence that thiazolidinediones (TZDs) such as pioglitazone and rosiglitazone, used in the treatment of T2D, increase subcutaneous adipocytes' storage capacity and lower T2D risk[60]. Results of the VICTORY trial, a study aimed at assessing the safety and efficiency of rosiglitazone on saphenous vein graft atherosclerosis and the cardiometabolic risk profile, showed that rosiglitazone treatment induced a 3 kg weight gain over 12 months and no change in visceral adiposity[61]. Pioglitazone has also been shown to reduce hepatic steatosis and inflammation in patients with NASH[62] thereby providing randomized clinical trial support to our MR findings. Semaglutide, a glucagon-like protein-1 (GLP-1) receptor agonist, has recently been shown to increase the rate of NASH resolution compared with placebo[63]. Recent studies on another GLP-1 receptor agonist liraglutide and a dual glucose-dependent insulinotropic polypeptide (GIP) and GLP-1 receptor agonist also recently provided evidence that this pathway may induce a preferential loss in visceral adipose tissue and liver fat accumulation[57,64].

An important strength of the current study is the use of the largest liver fat accumulation and NAFLD datasets available to date. Additionally, the use of MVMR enabled the estimation of the direct effect of closely related risk factors on cardiometabolic outcomes while mitigating bias from confounding and reverse causality compared to classic observational studies. Our study, however, has limitations. The NAFLD GWAS included ~8000 cases and ~750,000 controls, but the population prevalence of NAFLD has been estimated to 25%. Hence, it is probable that some controls could have been misclassified. While it is important to acknowledge this limitation, we believe that such misclassification could bias our results towards the null and underestimate the strength of the reported associations. These associations were also consistent when using liver fat accumulation measured in 32,858 individuals, which better represents population prevalence for this trait compared with NAFLD. In contrast to adiposity-related traits, few genetic instruments were available for NAFLD and liver fat when these traits were used as exposures, making the assessment of pleiotropy more challenging. Consequently, we used a more lenient $p$-value threshold when NAFLD was used as the study exposure, increasing variance explained with the drawback of having more chance of including invalid or pleiotropic instruments. Another potential limitation to

this work is that a binary trait (NAFLD) was used as an exposure. This could have led to the violation of the exchangeability assumption[44]. For this reason, we only tested the causal null hypothesis, instead of attempting to calculate the causal estimate. We also used the underlying continuous risk factor "liver fat accumulation" to estimate causal estimates. Finally, although the instrument strength was adequate to perform univariable MR analyses, waist circumference and BMI had low conditional F-statistics in MVMR, making these instruments vulnerable to weak instrument bias. Robust MR analyses and egger intercept indicated that other assumptions were likely to be satisfied.

In conclusion, results of this MVMR investigation suggest that independently of BMI, waist circumference is a strong and causal contributor to NAFLD. Also, the association between waist circumference and T2D and CAD is largely independent of liver fat. Altogether, the results are consistent with the hypothesis that abdominal adiposity may represent a root cause of cardiometabolic diseases. Clinical interventions targeting ectopic lipid deposition may be the key to the treatment of cardiometabolic diseases such as NAFLD, CAD, and T2D.

**Institutional review board approval**. All GWAS summary statistics were publicly available and accessible through URL. For all included genetic association studies, all participants provided informed consent and study protocols were approved by their respective local ethical committee.

## Data availability

Source data and GWAS summary statistics can be found following the following links: GWAS summary statistics for anthropometric traits from GIANT are available at: https://portals.broadinstitute.org/collaboration/giant/index.php/GIANT_consortium_data_files GWAS summary statistics for BMI from UKB are available via the MR Base GWAS catalog at id "ukb-b-19953". GWAS summary statistics for waist circumference from UKB are available via the MR Base GWAS catalog at id "ukb-b-9405". GWAS summary statistics for T2D are available at: http://diagramconsortium.org/downloads.html GWAS summary statistics for CAD are available at: https://www.cardiomics.net/download-data. GWAS summary statistics for NAFLD are available at: https://www.ebi.ac.uk/gwas/studies/GCST90091033. We make accessible a small subset of these summary statistics to reproduce the figures and the results on our GitHub[65].

## Code availability

The code used to perform the analysis can be found on GitHub[65]. The *TwoSampleMR* package is available at: https://github.com/MRCIEU/TwoSampleMR The *MendelianRandomization* package is available at: https://github.com/cran/MendelianRandomization The *data.table* package is available at https://github.com/Rdatatable/data.table The *tidyverse* package collection is available at: https://github.com/tidyverse/tidyverse.

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

## Acknowledgements

We would like to thank all study participants as well as all investigators of the studies that were used throughout the course of this investigation. E.G. and I.M.P. hold a doctoral research award from the *Fonds de recherche du Québec: Santé.* (FRQS). W.P. holds a masters research award from the Canadian Institutes of Health Research (CIHR). B.J.A. and S.T. hold junior scholar awards from the FRQS. M.C.V. is Canada Research Chair in Genomics applied to Nutrition and Metabolic Health. A.T. is co-Director of the Research Chair in Bariatric and Metabolic Surgery at Laval University. Part of this study was supported by the European Union through the European Regional Development fund. The work of Estonian Genome Center, Univ. of Tartu has been supported by the European Regional Development Fund and grants No. GENTRANSMED (2014-2020.4.01.15-0012), MOBERA5 (Norface Network project no 462.16.107), and 2014-2020.4.01.16-0125. This study was also funded by the European Union through Horizon 2020 research and innovation program under grant no 810,645 and through the European Regional Development Fund.

## Author contributions

Conceptualization and design E.G., W.P., and B.J.A. Data collection or analysis: E.G., W.P., N.T., E.A., T.E., N.G., J.B., I.M.P., É.G. Data Interpretation E.G., H.M., S.T., P.L.M., M.C.V., A.T., J.P.D., and B.J.A. Drafting of the manuscript E.G., W.P., B.J.A., and P.L.M. Critical review N.T., E.A., T.E., N.G., J.B., I.M.P., H.M., É.G., S.T., M.C.V., A.T., and J.P.D. All authors approve this version of the article and acknowledge their responsibility.

## Competing interests

The authors declare the following competing interests: B.J.A. is a consultant for Novartis and Silence Therapeutics and has received research contracts from Pfizer, Ionis Pharmaceuticals and Silence Therapeutics. A.T. receives research funding from Johnson & Johnson Medical Companies, Medtronic, Bodynov, and GI Windows for studies on bariatric surgery and received consulting fees from Novo Nordisk and Bausch Health. All other authors declare no competing interests.
