## [Peer Review File · Communications Medicine]

Reviewers' comments:

Reviewer #1 (Remarks to the Author):

In this study, the authors show that abdominal adipose tissue (WC) is the strongest adipose correlate of metabolic abnormalities which has been up to recently attributed to an excess amount of total body fat (BMI) using a Mendelian Randomization (MR) design. Furthermore, the authors debunk the popular hypothesis states that metabolic consequences of abdominal adiposity are mediated by excess liver fat using multivariable MR.

Waist circumference is shown as causal for NAFLD, because it's a better proxy for abdominal visceral adipose tissue (VAT). Neither WC nor BMI explain the causal mechanism by which body fat patterns affect health risk, and the current study merely scratched the surface and missed an opportunity to examine various pathophysiological mechanisms underlying the obesity - liverfat relationship, such as insulin resistance, hypertriglyceridemia [TG, apo-B], inflammation [TNFA, IL-6], adipokine dysregulation [adiponectin], and a disturbed hormonal milieu [testosterone, sex hormone binding globulin]. Furthermore, smokers have been shown to have more abdominal adipose tissue and to be characterized by more insulin resistance despite the fact that they tended to have lower BMI values than nonsmokers. An MVMR including these potential intermediates would foster some interesting results.

The authors instead chose for a questionable bidirectional MR and test whether NAFLD is a risk factor for BMI. Excess body fat is an obvious risk factor for increased liver fat, but I can't imagine how liver fat would be a risk factor for body weight. The only causal relationship that comes to mind is one of reverse causality, where patients with NAFLD are promoted to lose weight. A more relevant choice would be to study the bidirectional nature between NAFLD and T2D, as insulin resistance has a major role in the development of steatosis. But steatosis itself also promotes insulin resistance, endorsing a self-perpetuating vicious cycle. It has been reported that 55% of patients with T2D will develop NAFLD, as compared to 25% of the general population.

As the authors allude to in their discussion, it's been shown that increased intra-abdominal visceral 'metabolically active' adipose tissue (VAT) rather than subcutaneous adipose tissue (SAT) is causal for metabolic derangements, and WC, while a more accurate marker than BMI, does not distinguish between the two storage depots. Have the authors considered using direct instrument of visceral adipose tissue and abdominal subcutaneous adipose tissue (PMID 34128465)?

Recent methodological work used 159 adiposity-related genetic variants reflecting three adiposity traits (BMI, WHR, and WHR-adjusted BMI) to classify genetic instruments for adiposity subtypes classified according to differences in body fat distribution and metabolic features [PMID: 29769528]. It would be interesting to see whether genetic adiposity sub-types as instruments for MR analyses to examine the associations between genetically predicted unfavorable, favorable and neutral adiposity on the risk of NAFLD.

Minor:

Line 48: Waist and BMI are strongly correlated, and I suspect that they share many genetic variants, which may explain why their clinical outcomes are not markedly different when studies in isolation. Can the authors specifically state what the overlap is between BMI and WC loci?

Line 131: The use of WHR adjusted for BMI is discussed without any introduction or context.

Line 138: The authors should consider doing bidirectional analyses between NAFLD and T2D.

Figure 2: consider removing. Just have Figure 3, univariate estimates, but focus on MVMR.

Line 288: Misclassification of the exposure, 8,000 NAFLD cases and 700K controls, lead to a prevalence of 1% of NAFLD, whereas the true population prevalence of NAFLD is 25%. The control group probably therefore to contain many NAFLD subjects.

line 77: pretty sure there is literature out there on the causal relationship between WC and CAD and T2D.

Emdin CA, Khera AV, Natarajan P, et al. Genetic Association of Waist-to-Hip Ratio With Cardiometabolic Traits, Type 2 Diabetes, and Coronary Heart Disease. *JAMA*. 2017;317(6):626–634. doi:10.1001/jama.2016.21042

Chen Q, Li L, Yi J, et al. Waist circumference increases risk of coronary heart disease: Evidence from a Mendelian randomization study. *Mol Genet Genomic Med*. 2020;8(4):e1186. doi:10.1002/mgg3.1186

Li K, Feng T, Wang L, Chen Y, Zheng P, Pan P, Wang M, Binnay ITS, Wang Y, Chai R, Liu S, Li B, Yao Y. Causal associations of waist circumference and waist-to-hip ratio with type II diabetes mellitus: new evidence from Mendelian randomization. *Mol Genet Genomics*. 2021 May;296(3):605-613. doi: 10.1007/s00438-020-01752-z. Epub 2021 Feb 25. PMID: 33629185.

line 78-79: There are no FDA-approved targets aimed at targeting NAFLD?

Reviewer #2 (Remarks to the Author):

Gagnon et al. have conducted a Mendelian randomization study on obesity, non-alcoholic fatty liver disease (NAFLD) and cardiometabolic outcomes. The authors have made a comprehensive job on the analyses. I do have several comments to clarify some potential issues with the manuscript, and I hope that the comments will improve the analyses and their interpretations.

Major comments:

1. The authors have employed four genetic variants as instrumental variables for NAFLD liability. Such low number of variants is not optimal for the MR sensitivity analyses. As an example, the MR-Egger method estimates the intercept (the estimated "pleiotropic" effect) and the slope (the estimated "causal" effect) based on a regression of only four variants. The MR-Egger intercept results are not reported, but my educated guess is that (with NAFLD risk as the exposure) the confidence intervals will be very wide, and therefore cannot rule out even large pleiotropic effects. One approach would be to use a more lenient p-value threshold compared to the (more or less arbitrary) threshold of $p < 5e-8$. The limitations in employing such few number of instruments and the subsequent problems in the sensitivity analyses described above should be at least thoroughly

discussed.

2. The MR results of NAFLD risk on CAD risk were inconclusive (see also other comment below). It is unclear why NAFLD-CAD was then carried out for further MVMR analyses - it is expected that a result with no clear evidence for association in univariate MR is not going to show evidence for association in MVMR. Therefore, with these data, I am not convinced there is much to be learnt from the MVMR analysis on CAD risk.

Other comments:

3. Page 3, line 84: it is debatable whether Mendelian randomization (MR) is a "new" method per se, with perhaps the first proposition by Katan in 1986 and the seminal paper by Davey Smith and Ebrahim almost 20 years ago. A reference to any general introductory paper to MR would also be adequate.

4. Supplementary Tables:

Please provide column header descriptions and the abbreviations used, as well as odds ratios for binary outcomes.

ST1: some of the PMIDs seem to be missing; please separate cases and controls in the sample sizes.

ST2: please provide the MR-Egger intercept estimate, standard error and confidence interval.

5. Figure 2: please provide the effect sizes on NAFLD risk as odds ratios.

6. Methods and results throughout: how were R² and F-statistics calculated? Both cited papers (<https://doi.org/10.1093/ije/dyr036> and <https://doi.org/10.1093/ije/dyq151>) report the Cragg-Donald F-statistic, which itself is based on an estimate of R², but it is unclear how the R² was calculated for the multiple instruments based on summarised data. A common way to approximate this is the sum of the individual R² (as the stringent clumping provides reasonable independence of the variants), however I presume this was not the way they were calculated here. Please clarify.

7. Related to the previous comment: how were the R² and F-statistics calculated for the binary exposure? There are multiple ways to calculate this (<https://doi.org/10.1002/gepi.21614>).

8. Methods and results throughout: The genetic associations for binary phenotypes represent the (log) odds ratios, and they measure the liability to a trait, rather than its observed manifestation. Therefore, it would be more accurate to talk about "NAFLD risk" instead of "NAFLD". The same applies to T2DM and CAD.

9. Page 6, line 163 onwards: the authors report genetic liability to NAFLD "not associated with CAD [risk]" - however the 95% CI shows the data being consistent with an OR as low as 0.68, or as high as 1.24. A more appropriate interpretation would be to present the results as inconclusive. Please also see the major comment (2.) above.

10. Page 10, line 283: the "principal components" most likely refer to genetic principal components - please explicitly state this.

11. Page 10, line 283: was the waist circumference GWAS adjusted for height, and if not, how would this affect the results?

12. Page 11, lines 327-328: please give the exact coordinates and the genomic build for the genomic

regions excluded.

13. Page 11, line 330: what are "assumptions two and three"?

14. Throughout: please be more accurate with the units of the effect sizes - i.e. the effect size is per 1-unit (standard deviation?) increase in the genetically predicted exposure.

Reviewer #3 (Remarks to the Author):

In this paper, the authors set out to use Mendelian randomization to understand two things: 1) is BMI or WC the causal driver of NAFLD; 2) is BMI/WC or NAFLD the causal driver of T2D and CAD. This is nice work. The authors have included most of the things I want to see in an MR paper but there are some details missing and I have some questions regarding the methodology. The methods used (bidirectional MR followed by multivariable MR) are right for these questions and suitable sensitivity analyses are performed. I have a few major comments and some minor points for consideration/clarification.

Major points:

1. Please can the authors complete and provide a STROBE-MR checklist (<https://www.equator-network.org/reporting-guidelines/strobe-mr-statement/>).
2. Terminology – you use a number of terms that aren't appropriate to the question you are asking. Specifically, you use obesity, abdominal obesity, body weight and related terms. You are using genetic variants associated with, for example, variation in WC. You are not using a clinical definition of abdominal obesity which is equivalent to a WC of > XX. Consider using different terms (e.g. adiposity) throughout, including in the title.
3. Data and code. You have a section called "Data and code availability" but provide none of the data or code that you used. Can you (i) share all of the code used in this work (e.g. on GitHub) preferably with a doi (e.g. Zenodo); (ii) give date last accessed for data used (e.g. for DIAGRAM); (iii) provide direct links or explicit file names for GIANT GWAS data.
4. WHR adjusted for BMI – three things here:
 - a. Justification - it's not clear what the aim of this was nor whether it was a primary/secondary or sensitivity analysis (although I assume the latter since it is not mentioned in the 'aims' at the end of the introduction). The sentence in the results relating to this (p.5, L133-6) is difficult to understand.
 - b. consider whether this analysis is appropriate in the context of potential collider bias when instrumenting adjusted traits in this way – see: <https://academic.oup.com/ije/article/50/5/1639/6146681>
 - c. there are no instruments in supp table 4 and no stats in supp table 5 relating to this exposure, why? Also, there is no info in the results of the SNP N like you have for BMI and WC.
5. Workflow from one analysis to the next, including how decisions were made regarding what variables to take forward, is not clear. For example, it is not clear why WHRadjBMI was not taken forward to MVMR analysis, nor why NAFLD was taken forward as an exposure (alongside WC) in the MVMR with CAD as the outcome, even though univariable analysis showed no association between NAFLD and CAD. This needs to be clarified and reflected in Fig 1. This extends to other figures/tables, for instance Fig 2 has no estimates from WHRadjBMI on NAFLD. In addition, careful consideration needs to be given to analyses involving NAFLD and CAD given in univariable MR analysis there appears to be, if anything, a protective effect of NAFLD on CAD, but in multivariable MR the assumption is that NAFLD is mediating the effect of WC on CAD.
6. Justification for the use (or not) of additional/alternative GWAS for exposures and outcomes are not clear. For instance: (i) just using an additional exposure GWAS is not replication, especially if it's

a subset of what was used before (as in the case when you exclude UKBiobank); (ii) there is no ref for the BMI and WC UKB GWAS; (iii) what is the % overlap in the exposure and outcome when including UKB in exposure; (iv) why don't you use additional exposure GWAS for WHRadjBMI and NAFLD; (v) sample overlap only biases estimates in the presence of weak instruments which is not the case for BMI, WC, and WHRadjBMI (it may be for NAFLD) see - <https://www.medrxiv.org/content/10.1101/2021.06.28.21259622v1.full>. Clarity is needed re. which are the main analyses, and which are sensitivity analyses (currently you seem to use replication/sensitivity interchangeably).

7. Mediation analysis – I'm not sure that calculating the indirect effect as you do here is appropriate with a binary outcome given what Stephen Burgess says in the ref you provide "In practice, as in the applied example considered in this paper, we would recommend providing estimates of the total and direct effects, but not the indirect effect, as calculation of the indirect effect relies on the linearity of the relationships that cannot occur with a binary outcome." I suggest reading this paper <https://www.ncbi.nlm.nih.gov/pmc/articles/PMC8159796/>. In addition, the way you calculate the mediated effect here is using two different SNP lists, the power of these is therefore no longer the same and so presumably, any difference you see may be a consequence of that change in power?

Minor points:

8. Significance. I would discourage the use of the term "significant" throughout – see <https://www.ncbi.nlm.nih.gov/pmc/articles/PMC1119478/> for justification for this.

9. Try to be consistent with descriptions and the presentation of methods/results. For example: the GWAS descriptions are not all given the same level of detail (there is also information missing – see STROBE-MR). There are also some inconsistencies in how the different elements of the work are presented across methods and results making it difficult to follow in places. It might help to have the same headings for both so that they are reflective of one another (the same applies to Fig 1)?

10. Units presented across figures are not consistent (both OR and logOR used) although in the methods it says, 'odds ratios were reported'.

11. It is not clear whether the GWAS of NAFLD was performed in this work or previously, I suspect previously. Add "Previously, we performed...." on line 289 if the latter.

12. You present results both in SD units and then per unit increase. As I understand it, the GWAS you extract betas from involved analyses on inverse rank normal transformed (RNT) data. Whilst this will give SD units, it's not the same as SD units from just z-scoring (i.e., centering and scaling) and as such, caution is needed when trying to 'back transform' any estimates to original units. My experience with this is that such back-transformation only really works if the trait distribution was close to normal in the first place. Whilst I appreciate your efforts to present the results in a more interpretable way (by giving unit increases), I think you need to be clear how this back transformation has been performed and the assumptions underpinning it (including the assumed trait SD). Also you say: "... residuals were transformed to approximate normality with SD of 1 using inverse normal scores.", but assuming ties are dealt with correctly, an RNT should force a perfect normal distribution.

13. Line 149-151, you say 24% mediated. Then line 170 you say "modestly". At a population level I imagine 24% would be a large reduction in T2D cases(?). Please contextualise (in terms of incidence/risk) if possible (although also see comment above re the mediation analysis).

14. Discussion – last two sentences "Altogether....". I don't think your results say subcutaneous and visceral fat represent a root cause of a broad range of cardiometabolic traits. You investigated T2D and CAD, that's not a broad range of traits. You also didn't look at subcutaneous and visceral fat, you looked at BMI and WC, there are lots of other measures and specific SAT/VAT GWAS data is available. I think this concluding para needs re-writing.

15. In the discussion on line 252 you say NAFLD isn't associated with T2D but in the results you say it is and in figure 5 it looks like there's an association.
16. Introduction - lines 84-91 – suggest adding a reference for MR. e.g., <https://pubmed.ncbi.nlm.nih.gov/12689998/>
17. Results - line 152 - how many SNPs for WC from GIANT? The reporting isn't consistent across the exposures in this section.
18. Discussion - line 208-212 – I think these lines are missing a reference
19. Discussion - line 243-245 - MVMR as a method isn't absent confounding, confounding in MR is a result of instruments/populations, you can't eliminate that just by using a multivariable approach. I think you should re-write this.
20. supp figures - it goes supp figure 1, 2, and then 4. where is supp figure 3? Should supp 4 actually be supp 3.
21. Supp tables 4 and 5 are missing info for WHRadjBMI
Typos, grammar, etc:
22. Results - line 147 - missing) after the p-value for WC it should be "...p=5.7e-45) and NAFLD"
23. Figure 1A – typo in UKBiobank in top bubble

Reviewer #1 (Remarks to the Author):

In this study, the authors show that abdominal adipose tissue (WC) is the strongest adipose correlate of metabolic abnormalities which has been up to recently attributed to an excess amount of total body fat (BMI) using a Mendelian Randomization (MR) design. Furthermore, the authors debunk the popular hypothesis states that metabolic consequences of abdominal adiposity are mediated by excess liver fat using multivariable MR.

Authors' response: *We would like to thank reviewer #1 for his/her valuable comments on our manuscript and for suggesting how it can be improved.*

1) Waist circumference is shown as causal for NAFLD, because it's a better proxy for abdominal visceral adipose tissue (VAT). Neither WC nor BMI explain the causal mechanism by which body fat patterns affect health risk, and the current study merely scratched the surface and missed an opportunity to examine various pathophysiological mechanisms underlying the obesity - liverfat relationship, such as insulin resistance, hypertriglyceridemia [TG, apo-B], inflammation [TNFA, IL-6], adipokine dysregulation [adiponectin], and a disturbed hormonal milieu [testosterone, sex hormone binding globulin]. Furthermore, smokers have been shown to have more abdominal adipose tissue and to be characterized by more insulin resistance despite the fact that they tended to have lower BMI values than nonsmokers. An MVMR including these potential intermediates would foster some interesting results.

Authors' response: *This is an excellent point. We agree with that evaluating the question: "what mediates the causal effect of waist circumference on NAFLD?" in a multivariable MR framework is a very important research question. We have performed some of the additional analyses suggested by the reviewer to identify some factors that may mediate this association. However, the interpretation of*

these results was not as straightforward as we would have hoped and felt that these actually generated more questions than they answered. Additionally, since all reviewers asked us to perform an important number of additional analyses to consolidate our original findings, we focused our efforts towards this end and we are very pleased to provide a revised version of the manuscript that better disentangles the relationships between abdominal adiposity, liver fat/NAFLD and cardiometabolic diseases.

The authors instead chose for a questionable bidirectional MR and test whether NAFLD is a risk factor for BMI. Excess body fat is an obvious risk factor for increased liver fat, but I can't imagine how liver fat would be a risk factor for body weight. The only causal relationship that comes to mind is one of reverse causality, where patients with NAFLD are promoted to lose weight. A more relevant choice would be to study the bidirectional nature between NAFLD and T2D, as insulin resistance has a major role in the development of steatosis. But steatosis itself also promotes insulin resistance, endorsing a self-perpetuating vicious cycle. It has been reported that 55% of patients with T2D will develop NAFLD, as compared to 25% of the general population.

Authors' response: *We agree and removed the assessment of the causal effect of NAFLD on anthropometric measurements in the main manuscript. Figure 2 has been modified accordingly. As suggested by this reviewer, the new analyses presented in the manuscript expand the analyses on the relationship between BMI, waist, liver fat and NAFLD. These new analyses revealed that T2D may indeed be a causal contributor to liver fat accumulation and NAFLD. This new information is now presented in Supplementary Table 3 and is now discussed in the manuscript. We'd like to thank the reviewer for this helpful suggestion.*

2) As the authors allude to in their discussion, it's been shown that increased intra-abdominal visceral 'metabolically active' adipose tissue (VAT) rather than subcutaneous adipose tissue (SAT) is causal for metabolic derangements, and

WC, while a more accurate marker than BMI, does not distinguish between the two storage depots. Have the authors considered using direct instrument of visceral adipose tissue and abdominal subcutaneous adipose tissue (PMID 34128465)?

Authors' response: *We thank the reviewer for this comment. We agree that evaluating the direct effect of different organs fat content in a multivariable MR framework would be very enlightening. While performing additional analyses to address this point, we noticed that there are only two genome-wide significant independent genetic instruments for ASAT and that there are only five genetic instruments available for VAT. Hence, we could not perform this analysis. We, however, included liver fat from the same data source suggested. Although we agree with the reviewer that investigating VAT and ASAT could have been a valuable addition to our analyses, VAT and ASAT are highly correlated with BMI so any variant associated with either VAT or ASAT could be a consequence of excess total adiposity. Additionally, there is currently a debate in the literature as to whether dysfunctional SAT or the absolute amount of VAT is the driving force behind the metabolic consequences of impaired body fat distribution. Consequently, we have opted to study WHRadjBMI, which we believe is the best measure to investigate the metabolic consequences of impaired body fat distribution. We would also like to point at this point that we have performed new analyses to investigate this research question (see our response to the third comment from this reviewer just below).*

3) Recent methodological work used 159 adiposity-related genetic variants reflecting three adiposity traits (BMI, WHR, and WHR-adjusted BMI) to classify genetic instruments for adiposity subtypes classified according to differences in body fat distribution and metabolic features [PMID: 29769528]. It would be interesting to see whether genetic adiposity sub-types as instruments for MR analyses to examine the

associations between genetically predicted unfavorable, favorable and neutral adiposity on the risk of NAFLD.

Authors' response: *We'd like to thank the reviewer for suggesting this very interesting paper showing that a GRS including SNPs that are positively associated with BMI but negatively associated with WHR was associated with a favourable metabolic profile, particularly decreased risk of T2D and CAD. We performed this analysis adding NAFLD as the outcome. This analysis provided further evidence of a role of abdominal adiposity in the etiology of NAFLD that is remarkably consistent with the MVMR results showing that showing differential effect of BMI on NAFLD depending of the class of instruments used. These new results are presented in a new figure (Figure 3).*

Line 48: Waist and BMI are strongly correlated, and I suspect that they share many genetic variants, which may explain why their clinical outcomes are not markedly different when studies in isolation. Can the authors specifically state what the overlap is between BMI and WC loci?

Authors' response: *We agree with this comment. BMI and waist circumference shared 144 SNPs, which may explain why, in univariable analyses, they yield similar estimates. This information was included in the third paragraph of the Results section.*

Line 131: The use of WHR adjusted for BMI is discussed without any introduction or context.

Authors' response: *Good point. We now provide a clearer explanation as to why this variable is of interest in the context of this paper and why it was used in the second paragraph of the Results section of the revised manuscript.*

Line 138: The authors should consider doing bidirectional analyses between NAFLD and T2D.

Authors' response: *We agree and included the results of this analysis in Supplementary Table 3. This analysis revealed a causal contribution of T2D on NAFLD.*

Figure 2: consider removing. Just have Figure 3, univariate estimates, but focus on MVMR.

Authors' response: *We agree and now provide a modified Figure 2 (now Figure 1) that include the univariable analysis on the causal effect of WC, BMI and WHRadjBMI on NAFLD.*

Line 288: Misclassification of the exposure, 8,000 NAFLD cases and 700K controls, lead to a prevalence of 1% of NAFLD, whereas the true population prevalence of NAFLD is 25%. The control group probably therefore to contain many NAFLD subjects.

Authors' response: *This is an important point. NAFLD is unfortunately an underdiagnosed condition. We discussed this aspect in the limitation section of the Discussion. While it is important to acknowledge this limitation, we believe that such “misclassification” could bias our results towards the null and underestimate the strength of the associations that were observed throughout this manuscript.*

line 77: pretty sure there is literature out there on the causal relationship between WC and CAD and T2D.

Emdin CA, Khera AV, Natarajan P, et al. Genetic Association of Waist-to-Hip Ratio With Cardiometabolic Traits, Type 2 Diabetes, and Coronary Heart Disease. *JAMA*. 2017;317(6):626–634. doi:10.1001/jama.2016.21042

Chen Q, Li L, Yi J, et al. Waist circumference increases risk of coronary heart disease: Evidence from a Mendelian randomization study. *Mol Genet Genomic Med*. 2020;8(4):e1186. doi:10.1002/mgg3.1186

Li K, Feng T, Wang L, Chen Y, Zheng P, Pan P, Wang M, Binnay ITS, Wang Y, Chai R, Liu S, Li B, Yao Y. Causal associations of waist circumference and waist-to-hip ratio with type II diabetes mellitus: new evidence from Mendelian randomization. *Mol Genet Genomics*. 2021 May;296(3):605-613. doi: 10.1007/s00438-020-01752-z. Epub 2021 Feb 25. PMID: 33629185.

Authors' response: *We agree there is plenty of MR literature on the causal role of adiposity on CAD and T2D, but less on the causal effect of waist on NAFLD/liver fat and the bidirectional relationships between NAFLD/liver fat and CAD/T2D. We modified line 77 to specify that the effect of liver fat on cardiometabolic diseases has rarely been investigated.*

line 78-79: There are no FDA-approved targets aimed at targeting NAFLD?

Authors' response: *NAFLD's first line treatment is usually weight loss through a combination of healthy eating and exercise. Some drugs that are used for the prevention and/or treatment of T2D such as GLP1R agonists, PPAR gamma agonists or SGLT2 inhibitors may have positive effects on liver fat accumulation/NAFLD. Interestingly, we now report a causal effect of T2D on NAFLD and liver fat in support of these findings. There is no FDA drug specifically aimed at decreasing NAFLD risk.*

Reviewer #2 (Remarks to the Author):

Gagnon et al. have conducted a Mendelian randomization study on obesity, non-alcoholic fatty liver disease (NAFLD) and cardiometabolic outcomes. The authors have made a comprehensive job on the analyses. I do have several comments to clarify some potential issues with the manuscript, and I hope that the comments will improve the analyses and their interpretations.

Authors' response: *We would like to thank reviewer #2 for his/her valuable feedback and comments on our manuscript. We are also grateful for the helpful suggestions that we hope, will improve the clarity of our manuscript.*

Major comments:

1. The authors have employed four genetic variants as instrumental variables for NAFLD liability. Such low number of variants is not optimal for the MR sensitivity analyses. As an example, the MR-Egger method estimates the intercept (the estimated "pleiotropic" effect) and the slope (the estimated "causal" effect) based on a regression of only four variants. The MR-Egger intercept results are not reported, but my educated guess is that (with NAFLD risk as the exposure) the confidence intervals will be very wide, and therefore cannot rule out even large pleiotropic effects. One approach would be to use a more lenient p-value threshold compared to the (more or less arbitrary) threshold of $p < 5e-8$. The limitations in employing such few number of instruments and the subsequent problems in the sensitivity analyses described above should be at least thoroughly discussed.

Authors' response: *We agree that four genetic instruments is a low number of SNPs to fully investigated pleiotropy. We performed new analyses using a more lenient p-value threshold at $5e-6$ (resulting in 12 genetic instruments). The results using these instruments were similar. These are presented in Supplementary Table*

7. In support of this, we also included the underlying continuous variable “liver fat” as exposure in our analyses, as described in our response to the comments of Reviewer #1. We’d like to thank the reviewer for giving us the opportunity to clarify this point.

2. The MR results of NAFLD risk on CAD risk were inconclusive (see also other comments below). It is unclear why NAFLD-CAD was then carried out for further MVMR analyses - it is expected that a result with no clear evidence for association in univariate MR is not going to show evidence for association in MVMR. Therefore, with these data, I am not convinced there is much to be learnt from the MVMR analysis on CAD risk.

- ***Authors’ response:*** *There is heterogeneity of the estimate linking NAFLD and CAD. One possible reason for this heterogeneity is that genetic instruments for NAFLD liability may underlie different biological mechanisms. Some SNPs are linked with obesity and adipose tissue distribution whereas some SNPs are linked with higher liver fat, but lower plasma triglycerides and LDL-C (Brouwers, Hepatol Comms, 2019). Therefore, we hypothesized that NAFLD may in fact be protective of CAD when controlling for WC. Nonetheless, we removed this MVMR analysis since mediation analysis where the indirect effect act in opposing direction of the total effect are prone to type I error (pmid: 33961203)*

Other comments:

3. Page 3, line 84: it is debatable whether Mendelian randomization (MR) is a "new" method per se, with perhaps the first proposition by Katan in 1986 and the seminal paper by Davey Smith and Ebrahim almost 20 years ago. A reference to any general introductory paper to MR would also be adequate.

Authors' response: *We agree and changed that sentence accordingly, removing the word “new”.*

4. Supplementary Tables:

Please provide column header descriptions and the abbreviations used, as well as odds ratios for binary outcomes.

Authors' response: *Column header descriptions, abbreviations and odds ratios for binary outcomes are now provided in the supplementary tables.*

ST1: some of the PMIDs seem to be missing; please separate cases and controls in the sample sizes.

Authors' response: *Thank you for pointing this out. The PMID are included in the table. The summary statistics were not published in a paper but came from a GWAS by the MRC-IEU as indicated in the table. We included cases and controls columns.*

ST2: please provide the MR-Egger intercept estimate, standard error and confidence interval.

Authors' response: *We included this information in the revised version of the manuscript in supplementary table 4.*

5. Figure 2: please provide the effect sizes on NAFLD risk as odds ratios.

Authors' response: *We agree. We now present odds ratio in the new figure (now figure 1).*

6. Methods and results throughout: how were R2 and F-statistics calculated? Both cited

papers (<https://doi.org/10.1093/ije/dyr036> and <https://doi.org/10.1093/ije/dyq151>) report the Cragg-Donald F-statistic, which itself is based on an estimate of R², but it is unclear how the R² was calculated for the multiple instruments based on summarised data. A common way to approximate this is the sum of the individual R² (as the stringent clumping provides reasonable independence of the variants), however I presume this was not the way they were calculated here. Please clarify.

Authors' response: *We now include the following information in the Methods: "We calculated R² for each individual SNPs. (...). We calculated the F statistics following this formula $F = \frac{n-k-1}{k} \left(\frac{R^2}{1-R^2} \right)$. Where n is the sample size, k is the number of instruments used and R² is the sum of the individual R² of each SNP."*

7. Related to the previous comment: how were the R² and F-statistics calculated for the binary exposure? There are multiple ways to calculate this (<https://doi.org/10.1002/gepi.21614>).

Authors' response: *We now include the following information at line 353 : "For binary exposures, we calculated R² using equation 10 in Lee et al., 2012 (Lee et al. 2012) used in the get_r_from_lor function in the TwoSampleMR"*

8. Methods and results throughout: The genetic associations for binary phenotypes represent the (log) odds ratios, and they measure the liability to a trait, rather than its observed manifestation. Therefore, it would be more accurate to talk about "NAFLD risk" instead of "NAFLD". The same applies to T2DM and CAD.

Authors' response: *We agree and made the change accordingly throughout the manuscript.*

9. Page 6, line 163 onwards: the authors report genetic liability to NAFLD "not associated with CAD [risk]" - however the 95% CI shows the data being consistent with an OR as low as 0.68, or as high as 1.24. A more appropriate interpretation would be to present the results as inconclusive. Please also see the major comment (2.) above.

Authors' response: *We agree and changed that sentence: "Using 10 SNPs ($r^2 = 0.04$; F -statistics = 165), a 1-SD increase in liver fat increased the risk of T2D (OR=1.26 95% CI=1.08-1.47, $p=3.8e-03$), (OR = 0.90, 95% CI=0.75-1.10, $p=3.0e-01$)."*

10. Page 10, line 283: the "principal components" most likely refer to genetic principal components - please explicitly state this.

Authors' response: *We agree and made the change accordingly in the fifth paragraph of the Results section: "Measures were corrected for age, age squared, sex, ancestry-based principal components and study sites."*

11. Page 10, line 283: was the waist circumference GWAS adjusted for height, and if not, how would this affect the results?

Authors' response: *The GWAS was not adjusted for height. Waist-to-height-ratio and waist-to-hips ratio have similar predictive value for NAFLD (Zheng et al. 2012) PMID: 22701476. It could be extrapolated that waist circumference and waist-to-height-ratio have similar predictive value for NAFLD.*

12. Page 11, lines 327-328: please give the exact coordinates and the genomic build for the genomic regions excluded.

Authors' response: *We agree and made the change accordingly in the “Selection of genetic variants and variants harmonization’s section of the Methods: “SNPs in a 2 Mb window of the HLA, ABO and APOE genetic regions were excluded due to their complex genetic architecture and their widespread pleiotropy (in GRCh37 6:28909037-30913661, 9:135130951- 137150617, and 19:44409011-46412650, respectively).”*

13. Page 11, line 330: what are "assumptions two and three"?

Authors' response: *We agree and explicitly name them. “Selection of genetic variants and variants harmonization’ section of the Methods: “Exclusion of pleiotropic genetic regions satisfies the exclusion restriction and the exchangeability assumptions of instrumental variable analyses”.*

14. Throughout: please be more accurate with the units of the effect sizes - i.e. the effect size is per 1-unit (standard deviation?) increase in the genetically predicted exposure.

Authors' response: *Results of continuous variable are always reported on a 1 standard deviation scale. Results of binary outcomes are always reported in odds ratio (OR).*

Reviewer #3 (Remarks to the Author):

In this paper, the authors set out to use Mendelian randomization to understand two things: 1) is BMI or WC the causal driver of NAFLD; 2) is BMI/WC or NAFLD the causal driver of T2D and CAD. This is nice work. The authors have included most of the things I want to see in an MR paper but there are some details missing and I have some questions regarding the methodology. The methods used (bidirectional MR followed by

multivariable MR) are right for these questions and suitable sensitivity analyses are performed. I have a few major comments and some minor points for consideration/clarification.

Authors' response: *We would like to thank reviewer #3 for his/her valuable comments on our manuscript and for highlighted some important points that we hope will clarify our methods and help strengthen our conclusions.*

Major points:

1. Please can the authors complete and provide a STROBE-MR checklist (<https://www.equator-network.org/reporting-guidelines/strobe-mr-statement/>).

Authors' response: *We strongly agree. Initially, the submission system did not allow us to provide a STROBE-MR checklist. We provide it in the revised version of the manuscript.*

2. Terminology – you use a number of terms that aren't appropriate to the question you are asking. Specifically, you use obesity, abdominal obesity, body weight and related terms. You are using genetic variants associated with, for example, variation in WC. You are not using a clinical definition of abdominal obesity which is equivalent to a WC of > XX. Consider using different terms (e.g. adiposity) throughout, including in the title.

Authors' response: *We agree and replaced “obesity” with “adiposity” throughout the revised version of the manuscript.*

3. Data and code. You have a section called "Data and code availability" but provide none of the data or code that you used. Can you (i) share all of the code used in this work (e.g. on GitHub) preferably with a doi (e.g. Zenodo); (ii) give date last accessed for data used (e.g. for DIAGRAM); (iii) provide direct links or explicit file names for GIANT GWAS data.

Authors' response: *We agree. (i) You can access the code on https://github.com/gagelo01/replication_will_clean/Analysis (ii) Download dates are now provided in the supplementary table 1 (iii) We now provide url in the supplementary table 1.*

4. WHR adjusted for BMI – three things here:

a. Justification - it's not clear what the aim of this was nor whether it was a primary/secondary or sensitivity analysis (although I assume the latter since it is not mentioned in the 'aims' at the end of the introduction). The sentence in the results relating to this (p.5, L133-6) is difficult to understand.

b. consider whether this analysis is appropriate in the context of potential collider bias when instrumenting adjusted traits in this way –

see: <https://academic.oup.com/ije/article/50/5/1639/6146681>

c. there are no instruments in supp table 4 and no stats in supp table 5 relating to this exposure, why? Also, there is no info in the results of the SNP N like you have for BMI and WC.

Authors' response: *We agree.*

A. We changed the paragraph to better explain our rationale to include whradjbmi. Additionally, we made the last sentence clearer.

B. You are correct to point out that collider bias occurs in MR when adjusting for a child of the exposure. However, in our particular setting, we are interested in the effect of WHRadjBMI as a whole because it is a marker of fat distribution, not only WHR. The causal effect of WHRadjBMI is not adjusted for other covariate than sex, age, batch and genetic principal components which does not lead to collider bias. Additionally, this analysis is one of three methods that provide consistent evidence of an effect of intra-abdominal/visceral fat on liver fat accumulation/NAFLD.

C. Thank you for pointing that out. That was a mistake on our end. We now include WHRadjBMI in supplementary tables 4 and 5.

5. Workflow from one analysis to the next, including how decisions were made regarding what variables to take forward, is not clear. For example, it is not clear why WHRadjBMI was not taken forward to MVMR analysis, nor why NAFLD was taken forward as an exposure (alongside WC) in the MVMR with CAD as the outcome, even though univariable analysis showed no association between NAFLD and CAD. This needs to be clarified and reflected in Fig 1. This extends to other figures/tables, for instance Fig 2 has no estimates from WHRadjBMI on NAFLD. In addition, careful consideration needs to be given to analyses involving NAFLD and CAD given in univariable MR analysis there appears to be, if anything, a protective effect of NAFLD on CAD, but in multivariable MR the assumption is that NAFLD is mediating the effect of WC on CAD.

Authors' response: *We agree.*

1 - MVMR is a useful method when study exposures are correlated. In our case, WHRadjBMI is independent of BMI, as we know state at line 121-122. Hence, including WHRadjBMI in MVMR adds little value.

2- There is heterogeneity of the estimate linking NAFLD and CAD. One possible reason for this heterogeneity is that genetic instruments for NAFLD liability may underlie different biological mechanisms. Some SNPs are linked with obesity and adipose tissue distribution whereas some SNPs are linked with higher liver fat, but lower plasma triglycerides and LDL-C (Brouwers, Hepatol Comms, 2019, PMID: 30976747). Therefore, we hypothesized that NAFLD may in fact be protective of CAD when controlling for WC. Nonetheless, as suggested by reviewer #2, we removed this MVMR analysis since mediation analyses where the indirect effect act in opposing direction of the total effect are prone to type 1 error (pmid: 33961203).

3- Although Figure 1 was removed, the revised version of the manuscript better describes the workflow of our investigation.

6. Justification for the use (or not) of additional/alternative GWAS for exposures and

outcomes are not clear. For instance (i) just using an additional exposure GWAS is not replication, especially if it's a subset of what was used before (as in the case when you exclude UKBiobank); (ii) there is no ref for the BMI and WC UKB GWAS; (iii) what is the % overlap in the exposure and outcome when including UKB in exposure; (iv) why don't you use additional exposure GWAS for WHRadjBMI and NAFLD; (v) sample overlap only biases estimates in the presence of weak instruments which is not the case for BMI, WC, and WHRadjBMI (it may be for NAFLD) see - <https://www.medrxiv.org/content/10.1101/2021.06.28.21259622v1.full>. (vi) Clarity is needed re. Which are the main analyses, and which are sensitivity analyses (currently you seem to use replication/sensitivity interchangeably).

Authors' response: *We agree.*

- (i) We do not call it a replication dataset anymore.
- (ii) We now provide reference for BMI and WC UKB GWAS.
- (iii) 30% of the cases and 51% of the controls of the NAFLD GWAS came from the UK Biobank.
- (iv) Other independent GWAS summary statistics datasets for these two measures are not available to our knowledge.
- (v) We could not agree more, and we explicitly highlight this in the Methods. The instruments used to proxy NAFLD is strong ($r^2 = 0.0005$; F-statistics = 91).
- (vi) As the datasets we used are often not entirely independent, we dropped the use of the phrase "replication analyses."

7. Mediation analysis – I'm not sure that calculating the indirect effect as you do here is appropriate with a binary outcome given what Stephen Burgess says in the ref you provide "In practice, as in the applied example considered in this paper, we would recommend providing estimates of the total and direct effects, but not the indirect effect, as calculation of the indirect effect relies on the linearity of the relationships that cannot occur with a binary outcome." I suggest reading this paper <https://www.ncbi.nlm.nih.gov/pmc/articles/PMC8159796/>. In addition, the way you calculate the mediated effect here is using two different SNP lists, the power of these

is therefore no longer the same and so presumably, any difference you see may be a consequence of that change in power?

Authors' response: *We agree. We removed NAFLD as a mediator in MVMR analysis and used liver fat instead. The rationale is that liver fat is the underlying continuous risk factor of NAFLD. We thank you for the good reference on mediation analyses. The change in power should have little impact on the point estimate, but rather on the precision of the estimate. Despite including more instruments, MVMR estimates are less precise (as can be seen in Figure 2). Such large difference in point estimate between univariable and multivariable MR are most likely not due to the inclusion of more genetic instruments, but rather due to distinct direct effects.*

Minor points:

8. Significance. I would discourage the use of the term “significant” throughout – see <https://www.ncbi.nlm.nih.gov/pmc/articles/PMC1119478/> for justification for this.

Authors' response: *We agree and changed this section accordingly. “When BMI and waist circumference were assessed together in MVMR, only waist circumference (OR=2.35 95% CI=1.31-4.22, p=4.1e-03) retained a robust association with NAFLD, while the effect of BMI was inconclusive (OR=0.87 95% CI=0.54-1.38, p=5.5e-01) (Figure 2).”*

9. Try to be consistent with descriptions and the presentation of methods/results. For example, the GWAS descriptions are not all given the same level of detail (there is also information missing – see STROBE-MR). There are also some inconsistencies in how the different elements of the work are presented across methods and results making it difficult to follow in places. It might help to have the same headings for both so that they are reflective of one another (the same applies to Fig 1)?

Authors' response: *We agree with this point. In the revised version of the manuscript, we now provide much more information in Supplementary Table 1. This table describes all the cohorts used in this MR study. STROBE-MR guidelines checklist is also provided. In the text, we aimed at providing information that is not redundant to what can be found in Supplementary Table 1. We have also adapted the order in which we present the results and hope it is now clearer.*

10. Units presented across figures are not consistent (both OR and logOR used) although in the methods it says, 'odds ratios were reported'.

Authors' response: *We agree. $\log(OR)$ was used when a binary trait was used as exposures. OR were used when a binary trait is used as outcome. Since we no longer present estimates when using binary traits as exposures, only OR are presented when presenting a dichotomous outcome.*

11. It is not clear whether the GWAS of NAFLD was performed in this work or previously, I suspect previously. Add "Previously, we performed..." on line 289 if the latter.

Authors' response: *Indeed, the GWAS was performed in a previous investigation from our group. We changed this section accordingly : "Previously, we performed a GWAS meta-analysis for clinical diagnosis of NAFLD (8434 cases and 770,180 controls) of European ancestry from four cohorts (Ghodsian, 2021)."*

12. You present results both in SD units and then per unit increase. As I understand it, the GWAS you extract betas from involved analyses on inverse rank normal transformed (RNT) data. Whilst this will give SD units, it's not the same as SD units from just z-scoring (i.e., centering and scaling) and as such, caution is needed when trying to 'back

transform' any estimates to original units. My experience with this is that such back-transformation only really works if the trait distribution was close to normal in the first place. Whilst I appreciate your efforts to present the results in a more interpretable way (by giving unit increases), I think you need to be clear how this back transformation has been performed and the assumptions underpinning it (including the assumed trait SD). Also you say: "... residuals were transformed to approximate normality with SD of 1 using inverse normal scores.", but assuming ties are dealt with correctly, an RNT should force a perfect normal distribution.

Authors' response: *We strongly agree. We removed the results of BMI and WC using BMI points and cm respectively. We also modified this section to remove the word "approximate."*

13. Line 149-151, you say 24% mediated. Then line 170 you say "modestly". At a population level I imagine 24% would be a large reduction in T2D cases(?). Please contextualise (in terms of incidence/risk) if possible (although also see comment above re the mediation analysis).

Authors' response: *As suggested in comment 7, we removed the mediation analysis using NAFLD as a mediator. We instead used liver fat as a mediator. The new mediation results show that the effect of WC on T2D is 9% mediated by liver fat. To better understand how WC is a more important risk factor than liver fat we now state explicitly: "In MVMR, the effect of WC on T2D is 4.44 times larger than the effect of liver fat on T2D."*

14. Discussion – last two sentences "Altogether...". I don't think your results say subcutaneous and visceral fat represent a root cause of a broad range of cardiometabolic traits. You investigated T2D and CAD, that's not a broad range of traits. You also didn't look at subcutaneous and visceral fat, you looked at BMI and WC, there are lots of other

measures and specific SAT/VAT GWAS data is available. I think this concluding para needs re-writing.

Authors' response: *We agree. We rewrote this sentence: “the results are consistent with the hypothesis that abdominal adiposity may represent a root cause of cardiometabolic diseases. Clinical interventions targeting ectopic lipid deposition may be the key to the treatment of cardiometabolic diseases such as NAFLD, CAD and T2D.”*

15. In the discussion on line 252 you say NAFLD isn't associated with T2D but in the results you say it is and in figure 5 it looks like there's an association.

Authors' response: *Thank you for pointing that out it was an error on our end. We removed the phrase “T2D and” in the revised version.*

16. Introduction - lines 84-91 – suggest adding a reference for MR.

e.g., <https://pubmed.ncbi.nlm.nih.gov/12689998/>

Authors' response: *We added the recommended reference line 84-91.*

17. Results - line 152 - how many SNPs for WC from GIANT? The reporting isn't consistent across the exposures in this section.

Authors' response: *This information can be retrieved in supplementary table 5 and supplementary figures 1-2. The reporting is not consistent, because the inclusion of GIANT was used as a sensitivity analysis.*

18. Discussion - lines 208-212 – I think these lines are missing a reference

Authors' response: *We added references to support that statement.*

19. Discussion - lines 243-245 - MVMR as a method isn't absent confounding, confounding in MR is a result of instruments/populations, you can't eliminate that just by using a multivariable approach. I think you should re-write this.

Authors' response: *We agree and rewrote this section. "Additionally, the use of MVMR enabled the estimation of the direct effect of closely related risk factors on cardiometabolic outcomes while mitigating bias from confounding and reverse causality compared to classic observational studies."*

20. supp figures - it goes supp figure 1, 2, and then 4. where is supp figure 3? Should supp 4 actually be supp 3.

Authors' response: *Thank you for pointing this out that was a mistake on our end. There are only three supplementary figures.*

21. Supp tables 4 and 5 are missing info for WHRadjBMI

Typos, grammar, etc:

Authors' response: *We included info on WHRadjBMI in supplementary tables 3 and 4.*

22. Results - line 147 - missing) after the p-value for WC it should be "...p=5.7e-45) and NAFLD"

Authors' response: *Thank you. We added the parenthesis in the revised version.*

23. Figure 1A – typo in UKBiobank in top bubble

Authors' response: *Thank you. We removed figure 1 in the new version.*

Reviewers' comments:

Reviewer #1 (Remarks to the Author):

The authors have addressed my raised concerns with great precision. I have no further comments.

Reviewer #2 (Remarks to the Author):

Many thanks for the authors by taking the earlier reviewer comments into account. However, there remains a few points worth amending, which I believe would be of benefit for this manuscript.

Throughout:

As absence of evidence is not evidence of absence, I would expect some caution in the interpretation of results where the confidence interval overlaps the null. As an example, the authors write in the abstract: "BMI was not associated with NAFLD when accounting for waist circumference (0.87 95% CI=0.54-1.38, $p=5.5e-01$)". However, data are consistent with an OR as low as 0.54 or as high as 1.38 - therefore, "inconclusive" or "no clear evidence for association" would be more appropriate ways to describe the results here. Other examples include (this is not an exhaustive list): Page 6, line 164: "NAFLD caused T2D, but not CAD"; line 175: "CAD was not causally associated with liver fat accumulation and NAFLD" (ST3 actually shows some evidence for these associations) - please revise such statements throughout.

Regarding NAFLD as an exposure, the authors now present the results for 12 variants as instruments, however I would still expect some discussions of the potential difficulties in estimating evidence for pleiotropy with such few variants.

The authors write in the discussion: "although the instrument strength was adequate to perform univariable MR analyses, the use of MVMR might reduce instrument strength potentially leading to weak instrument bias". There indeed do exist F-statistics for MVMR (<https://doi.org/10.1093/ije/dyaa101>), which could be applied here.

F-statistics for NAFLD as an exposure: for the sample size in the F-statistic formula, was this the total sample size or the effective sample size? The latter is more meaningful for a binary outcome, especially in the presence of such imbalance between cases and controls as is the case here, and it is not completely clear which one is used here.

ST6: what are "ieu-a-835" or "ukb-b-19953"?

ST7: what is "ieu-a-7"? In addition, apologies but I could not find the corresponding MR-Egger intercept estimates in this table.

Reviewer #3 (Remarks to the Author):

The authors have done a good job of addressing mine and the other reviewers' comments and this is much improved. I also want to say they have done a good job with uploading their scripts to GitHub,

there is some organising and tidying needed to make it easily accessible but it is good for now. I have noted below a few minor points for the authors consideration. Line/page numbers refer to the PDF document.

Title

1. I think just Mendelian randomization is sufficient here given MVMR was not the sole MR analysis
- #### Abstract
2. Line 38 – add MVMR abbreviation here at first use
 3. Line 41 – $n >$ should really be N up-to XXX (this is the same for all exposure GWAS given not all SNPs may be present for all individuals)
 4. Line 46 – you say univariable MR but don't say this in the methods, on line 38 you just say MVMR – methods and results should correspond
 5. Line 53 – use WC not abdominal adiposity here, this is the key take away and it should be explicit regarding the measure used, if just glancing at the abstract this could be misinterpreted as WHRadjBMI or HC or VAT/SAT.

Introduction

6. Line 87 – Why do you mention WHRadjBMI here given the focus of the paper is on WC and BMI?
7. Line 95 – MVMR is not solely used when multiple genetic variants are associated with two or more exposures. Rephrase to “MVMR can be used....”
8. Line 101 – abbreviate MVMR
9. Last para of intro – nice, but why is there no mention of WHRadjBMI here?

Methods

10. Why are there two BMI/WC GWAS used while only one WHRadjBMI GWAS? What is the point of 'replication' if you are selective in what you replicate? There are multiple WHRadjBMI GWAS publicly available, so it is possible to do this.
11. Why does the WHRadjBMI GWAS still have such little description compared to all the other GWAS you use? E.g., I know the trait was inverse rank normally transformed prior to genome-wide analysis but this isn't stated in your description. There is also no information about the ancestry of this population. If you use the same wording for each dataset this will make reporting each dataset much more consistent.
12. Line 317-324 – liver fat was inverse rank normally transformed prior to genome-wide analysis, this needs to be included in your description.

Results

13. Line 110-111 - I'd really like a table that is just the exposure instruments to be honest, then I and anyone else can easily replicate your work, as opposed to a harmonized data frame
14. Line 109 – change the heading order so that NAFLD is before liver fat accumulation as this is the order it appears in the text
15. Line 128 – missing '.' after MR methods
16. WHRadjBMI is associated with NAFLD and liver fat but isn't included in MVMR analyses. If you are including WHRadjBMI and saying (in your response) that collider bias isn't an issue, then why not include it in MVMR?
17. Line 143-156 – I understand these additional analyses are a result of a reviewer comment, these have been done well and I think the interpretation here is good.
18. Line 163 – you imply that data for CAD are presented in figure 4 but this figure only has info for T2D.
19. Line 166 – you use a more lenient p-value threshold but give no commentary on this regarding the MR assumptions, i.e., the risk of doing this.

Discussion

20. L202 – typo ‘that’ instead of ‘than’

21. L212 – check for typo – sentence doesn’t seem to make sense. Plus, you talk about ‘body weight’ which is not investigated here per se (rather BMI).

Tables

22. The table titles and caption on the first sheet of the excel file are not detailed enough. Every table should be able to be read and understood outside of reading the manuscript. This means all columns need to be explained and so do abbreviations. E.g., what is ‘OR’ or ‘b’, what is column “pos.exposure” in Supp Table 2. What does “TRUE” mean in column “mr_keep”? What does “1” mean in column “action”. I understand this is a lot of work and is tedious but if you’re presenting all of this data (which you should and is great!) then it needs to be explained so people who haven’t used the TwoSampleMR and MVMR packages understand. Supp Table 8 is labelled as instrument strength for univariable MR but also includes heterogeneity stats by the look of it so the above applies to every table in the supp.

23. Supp Table 1 – what is “Fat_liver”? Assume this is liver fat, make sure consistent with wording in the main text.

24. Supp Table 2 – it is difficult to work out what SNPs are from which GWAS in Supp. Table 1. I would make another column for Table 1 with a trait ID e.g., trait_author_year_XXX where XXX is a distinction between GWAS from the same publication. This can then be used in subsequent Tables to uniquely identify phenotypes.

Figures

25. Figure 1 – typo in legend ‘inanthropometric’

26. Figure 4 – this only has data for T2D but in the results it’s implied that it has CAD results too

27. Figure 5 – mentioned ‘body weight’ – this is confusing as weight per se is not investigated here – wording needs to be consistent with that in the text. Legend correctly refers to BMI.

Reviewers' comments:

Reviewer #1 (Remarks to the Author):

The authors have addressed my raised concerns with great precision. I have no further comments.

Reviewer #2 (Remarks to the Author):

Many thanks for the authors by taking the earlier reviewer comments into account. However, there remains a few points worth amending, which I believe would be of benefit for this manuscript.

Authors' response: We would like to thank the reviewer for taking the time to read and comment our manuscript for a second time and for his/her important comments that we have integrated in the new version of our manuscript. .

Throughout:

As absence of evidence is not evidence of absence, I would expect some caution in the interpretation of results where the confidence interval overlaps the null. As an example, the authors write in the abstract: "BMI was not associated with NAFLD when accounting for waist circumference (0.87 95% CI=0.54-1.38, p=5.5e-01)". However, data are consistent with an OR as low as 0.54 or as high as 1.38 - therefore, "inconclusive" or "no clear evidence for association" would be more appropriate ways to describe the results here. Other examples include (this is not an exhaustive list): Page 6, line 164: "NAFLD caused T2D, but not CAD"; line 175: "CAD was not causally associated with liver fat accumulation and NAFLD" (ST3 actually shows some evidence for these associations) - please revise such statements throughout.

Authors' response: This is a very good point. We went through the entire manuscript once more and changed the wording in many sections so that we present our results more cautiously with a focus on effect sizes and confidence intervals rather than p-values.

Regarding NAFLD as an exposure, the authors now present the results for 12 variants as instruments. However I would still expect some discussions of the potential difficulties in estimating evidence for pleiotropy with such few variants.

Authors' response: We are not aware of any methodological papers showing that the difficulty of evaluating pleiotropy scales with the number of instruments. We added this

element of discussion in the limitations section of the manuscript: "In contrast to adiposity-related traits, few genetics instruments were available for NAFLD and liver fat when these traits were used as exposures, making the assessment of pleiotropy more challenging".

The authors write in the discussion: "although the instrument strength was adequate to perform univariable MR analyses, the use of MVMR might reduce instrument strength potentially leading to weak instrument bias". There indeed do exist F-statistics for MVMR (<https://doi.org/10.1093/ije/dyaa101>), which could be applied here.

Author's response: We thank the reviewer for this suggestion. We now include conditional F-statistics. Conditional F-statistics for this MVMR analysis were indeed low (Supplementary Table 5). Weak instruments make the results more vulnerable to horizontal pleiotropy bias. However, results were significant and consistent across all robust MVMR analyses and multivariable Egger intercept did not differ from zero indicating that pleiotropy is unlikely to have affected the results. This limitation was highlighted in the discussion.

F-statistics for NAFLD as an exposure: for the sample size in the F-statistic formula, was this the total sample size or the effective sample size? The latter is more meaningful for a binary outcome, especially in the presence of such imbalance between cases and controls as is the case here, and it is not completely clear which one is used here.

Author's response: We thank the reviewer for his thoughtful comment. Indeed, total sample size was used, which did not make much sense considering the case/control ratio imbalance. We now use effective sample size for binary exposures.

ST6: what are "ieu-a-835" or "ukb-b-19953"?

Authors' response: "ukb-b-19953" is BMI measured in the UK Biobank "ieu-a-835" is BMI derived in the GIANT consortium. We made sure to change the nomenclature for "BMI_UKB" and "GIANT_2015_BMI" respectively as it is the same nomenclature we use in Supplementary Table 1.

ST7: what is "ieu-a-7"? In addition, apologies but I could not find the corresponding MR-Egger intercept estimates in this table.

Author's response: "Ieu-a-7" is the coronary artery disease meta-analysis of GWAS performed by Nikpay et al. from the CARDIoGRAMplusC4D consortium. We made sure to change this for "Nikpay_CAD." We also included another supplementary table (Supplementary Table 8) to present Egger intercept for NAFLD with $p < 5e-6$.

Reviewer #3 (Remarks to the Author):

The authors have done a good job of addressing mine and the other reviewers' comments and this is much improved. I also want to say they have done a good job with uploading their scripts to GitHub, there is some organising and tidying needed to make it easily accessible but it is good for now. I have noted below a few minor points for the authors consideration. Line/page numbers refer to the PDF document.

Author's response: We would like to thank the reviewer for her/his constructive comments on how our manuscript could be improved. We would also like to extend our thanks to reviewer #4 who greatly helped in improving the manuscript.

Title

1. I think just Mendelian randomization is sufficient here given MVMR was not the sole MR analysis

Author's response: We made the change.

Abstract

2. Line 38 – add MVMR abbreviation here at first use

Author's response: We made the change.

3. Line 41 – $n >$ should really be N up-to XXX (this is the same for all exposure GWAS given not all SNPs may be present for all individuals)

Author's response: We agree that "N up to" could be more precise. However, we feel it just is a matter of definition. When describing the GWAS, we report the total sample size of the meta-analysis. Hence, we feel it is more adequate to use the "=" sign.

4. Line 46 – you say univariable MR but don't say this in the methods, on line 38 you just say MVMR – methods and results should correspond

Author's response: We agree and now use "univariable MR" instead of just "MR" when relevant throughout.

5. Line 53 – use WC not abdominal adiposity here, this is the key take away and it should be explicit regarding the measure used, if just glancing at the abstract this could be misinterpreted as WHRadjBMI or HC or VAT/SAT.

Authors' response: We agree and made the change.

Introduction

6. Line 87 – Why do you mention WHRadjBMI here given the focus of the paper is on WC and BMI?

Author's response: We used WHRadjBMI as a study exposure because it is an anthropometric measurement approximating fat distribution. WHRadjBMI gives us insight on the role of a body fat distribution pattern consistent with low peripheral/subcutaneous fat accumulation and high intra-abdominal fat accumulation. We clarified this in the introduction.

7. Line 95 – MVMR is not solely used when multiple genetic variants are associated with two or more exposures. Rephrase to “MVMR can be used....”

Author's response: We agree and made the change.

8. Line 101 – abbreviate MVMR

Author's response: We agree and made the change.

9. Last para of intro – nice, but why is there no mention of WHRadjBMI here?

Author's response: We agree and modified this sentence to include WHRadjBMI.

Methods

10. Why are there two BMI/WC GWAS used while only one WHRadjBMI GWAS? What is the point of ‘replication’ if you are selective in what you replicate? There are multiple WHRadjBMI GWAS publicly available, so it is possible to do this.

Authors' response: We were initially not aware of a WHRadjBMI GWAS in the UK Biobank. We thank you for pointing that these summary statistics existed in the public domain. We found one published by Pulit et al. 2019. We now include this in our analysis.

11. Why does the WHRadjBMi GWAS still have such little description compared to all the other GWAS you use? E.g., I know the trait was inverse rank normally transformed prior to genome-wide analysis but this isn't stated in your description. There is also no

information about the ancestry of this population. If you use the same wording for each dataset, this will make reporting each dataset much more consistent.

Author's response: Thank you for pointing this out: We included this information.

12. Line 317-324 – liver fat was inverse rank normally transformed prior to genome-wide analysis, this needs to be included in your description.

Author's response: Thank you. We included this information.

Results

13. Line 110-111 - I'd really like a table that is just the exposure instruments to be honest, then I and anyone else can easily replicate your work, as opposed to a harmonized data frame

Author's response: Recent MR guidelines suggest to report the harmonized data.frame to ensure the harmonization step was correct (Davies et al., 2018). The harmonized data.frame already contains all the information on the instruments.

14. Line 109 – change the heading order so that NAFLD is before liver fat accumulation as this is the order it appears in the text

Authors' response: This was done.

15. Line 128 – missing '.' after MR methods

Authors' response: A dot was added.

16. WHRadjBMI is associated with NAFLD and liver fat but isn't included in MVMR analyses. If you are including WHRadjBMI and saying (in your response) that collider bias isn't an issue, then why not include it in MVMR?

Author's response: We did not include WHRadjBMI in MVMR analyses because this exposure already includes waist circumference and BMI in its definition and we do not feel that adding these parameters in a MVMR analysis would be appropriate.

17. Line 143-156 – I understand these additional analyses are a result of a reviewer comment, these have been done well and I think the interpretation here is good.

Authors' response: We thank you for this encouraging comment.

18. Line 163 – you imply that data for CAD are presented in figure 4 but this figure only has info for T2D.

Author's response: We changed the sentence.

19. Line 166 – you use a more lenient p-value threshold but give no commentary on this regarding the MR assumptions, i.e., the risk of doing this.

Authors' response: We added a sentence commenting on the trade-off of using a more lenient p-value threshold.

Discussion

20. L202 – typo 'that' instead of 'than'

Authors' response: Thank you. We made the change.

21. L212 – check for typo – sentence doesn't seem to make sense. Plus, you talk about 'body weight' which is not investigated here per se (rather BMI).

Authors' response: Thank you. We changed "body weight" for BMI.

Tables

22. The table titles and caption on the first sheet of the excel file are not detailed enough. Every table should be able to be read and understood outside of reading the manuscript. This means all columns need to be explained and so do abbreviations. E.g., what is 'OR' or 'b', what is column "pos.exposure" in Supp Table 2. What does "TRUE" mean in column "mr_keep"? What does "1" mean in column "action". I understand this is a lot of work and is tedious but if you're presenting all of this data (which you should and is great!) then it needs to be explained so people who haven't used the TwoSampleMR and MVMR packages understand. Supp Table 8 is labelled as instrument strength for univariable MR but also includes heterogeneity stats by the look of it so the above applies to every table in the supp.

Authors' response: We agree with this point. We now provide on the first sheet of the excel supplementary table detailed information about the tables and their columns.

23. Supp Table 1 – what is "Fat_liver"? Assume this is liver fat, make sure consistent with wording in the main text.

Authors' response: Indeed, this is Liver fat. "Fat_liver" is the unique ID used in Supplementary Table 1.

24. Supp Table 2 – it is difficult to work out what SNPs are from which GWAS in Supp. Table 1. I would make another column for Table 1 with a trait ID e.g., trait_author_year_XXX where XXX is a distinction between GWAS from the same publication. This can then be used in subsequent Tables to uniquely identify phenotypes.

Authors' response: The column "trait" in table1 uniquely identifies exposures. The traits can be found in the column "exposure" and "outcome" in virtually all other supplementary tables. Following your comment #22 we made this clearer in the description of the supplementary tables.

Figures

25. Figure 1 – typo in legend 'inanthropometric'

Authors' response: We made the change.

26. Figure 4 – this only has data for T2D but in the results it's implied that it has CAD results too

Author's response: We made the change in the text to report that only results for T2D are reported in Figure 4.

27. Figure 5 – mentioned 'body weight' – this is confusing as weight per se is not investigated here – wording needs to be consistent with that in the text. Legend correctly refers to BMI.

Author's response: we changed "body weight" for "BMI".

Davies, N.M., Holmes, M.V., Davey Smith, G., 2018. Reading Mendelian randomisation studies: a guide, glossary, and checklist for clinicians. *BMJ* k601. <https://doi.org/10.1136/bmj.k601>

REVIEWERS' COMMENTS:

Reviewer #2 (Remarks to the Author):

Thanks for the authors for their revised manuscript. I have a couple of comments remaining:

Page 6, lines 154-155:

"negative ($p < 0.05$) association with WHR (BMI+WHR-) or null ($p < 0.05$) association with WHR (BMIonly+)"

Should the p-value inequality be the other way round for the null association?

Supplementary Table 7: what is "ieu-a-7"?

"We are not aware of any methodological papers showing that the difficulty of evaluating pleiotropy scales with the number of instruments"

The short comment in the discussion is now adequate. For the authors' interest, there is discussion on MR-Egger being, among other things, sensitive to outliers in Burgess & Thompson (2017):

<https://doi.org/10.1007/s10654-017-0255-x>

Such problems also follow trivially from elementary statistics: many of the MR sensitivity analyses (e.g. MR-Egger, MR-PRESSO) are regression-based methods, and e.g. for MR-Egger, there are three parameters (intercept, slope, residual σ^2) to be fit. The parameter estimates are bound to be very imprecise if there are only few data points available.

Reviewer #4 (Remarks to the Author):

I have no further comments.

Editorial requests. Please comment on the magnitude of conditional F statistics in the results section - currently only a reference is made to supplementary files in the results text.

Authors' response: We now comment on the magnitude of the conditional F-statistics for this analysis.

Line 140-144: Conditional F-statistics for this MVMR analysis were low (1.54 and 1.55 for WC and BMI respectively). However, results were significant and consistent across all robust MVMR analyses and multivariable Egger intercept did not differ from zero indicating that pleiotropy is unlikely to have affected the results (Supplementary Table 5).

Editorial request and also please change the heading/title of Figure 5 to an 'illustration of findings' or other similar general term, as these are not formal directed acyclic graphs (DAGs) (they are missing key features such as confounders, and one graph includes a bidirectional arrow, when DAGs should be acyclic/unidirectional only).

Author's response: We changed the figure title for "Schematic illustration of the main findings of the study."

Reviewer #2 (Remarks to the Author):

Thanks for the authors for their revised manuscript. I have a couple of comments remaining:

Page 6, lines 154-155:

"negative ($p < 0.05$) association with WHR (BMI+WHR-) or null ($p < 0.05$) association with WHR (BMIonly+) "Should the p-value inequality be the other way round for the null association?"

Authors' response: We thank you for pointing out this typo. This was modified in this new version.

Supplementary Table 7: what is "ieu-a-7"?

Authors' response: Thank you for pointing this out. "ieu-a-7" is the GWAS by Nickpay et al on coronary artery disease from the CARDIoGRAMplusC4D consortium excluding the UK Biobank. We changed this for "Nikpay_CAD" for consistency with the supplementary data.

"We are not aware of any methodological papers showing that the difficulty of evaluating pleiotropy scales with the number of instruments"

The short comment in the discussion is now adequate. For the authors' interest, there is discussion on MR-Egger being, among other things, sensitive to outliers in Burgess & Thompson (2017): <https://doi.org/10.1007/s10654-017-0255-x>

Such problems also follow trivially from elementary statistics: many of the MR sensitivity analyses (e.g. MR-Egger, MR-PRESSO) are regression-based methods, and e.g. for MR-Egger, there are three parameters (intercept, slope, residual σ^2) to be fit. The parameter estimates are bound to be very imprecise if there are only few data points available.

Author's response: Thank you for this very informative explanation!